# RNAinformer: Generative RNA Design with Tertiary Interactions

## Abstract

The function of an RNA molecule depends on its structure and a strong structure-to-function relationship is already achieved on the secondary structure level of RNA. Therefore, the secondary structure based design of RNAs is one of the major challenges in computational biology. A common approach to RNA design is inverse RNA folding. However, existing RNA design methods cannot invert all folding algorithms because they cannot represent all types of base interactions. In this work, we propose RNAinformer, a novel generative transformer based approach to the inverse RNA folding problem. Leveraging axial-attention, we directly model the secondary structure input represented as an adjacency matrix in a 2D latent space, which allows us to invert all existing secondary structure prediction algorithms. Consequently, RNAinformer is the first model capable of designing RNAs from secondary structures with all base interactions, including non-canonical base pairs and tertiary interactions like pseudoknots and base multiplets. We demonstrate RNAinformer's state-of-the-art performance across different RNA design benchmarks and showcase its novelty by inverting different RNA secondary structure prediction algorithms.

## 1 Introduction

Ribonucleic acid (RNA) is one of the major regulatory molecules inside the cells of living organisms with key roles during differentiation and development (Morris & Mattick, 2014). RNAs fold hierarchically (Tinoco Jr & Bustamante, 1999) and the structure is key to their function: Base interactions via hydrogen bonds result in a fast formation of a *secondary structure*, with tertiary interactions stabilizing the formation of the final 3D shape (Vicens & Kieft, 2022). A strong structure-to-function relationship is already achieved on a secondary structure level (Hammer et al., 2019), and therefore, RNA secondary structure prediction recently got into the focus of the deep learning community, achieving state-of-the-art results (Singh et al., 2019; Fu et al., 2022; Chen et al., 2022; Franke et al., 2022; 2024). Compared to more traditional methods, these algorithms predict an adjacency matrix representation of the secondary structure instead of the commonly used but less expressive dot-bracket string notation (Hofacker et al., 1994). This has the advantage that they are not limited to the prediction of specific kinds of base pairs but can model non-Watson-Crick interactions (Olson et al., 2019), pseudoknots (Staple & Butcher, 2005), as well as base multiplets (nucleotides that pair with more than one other nucleotide) (Bhattacharya et al., 2019; Singh et al., 2019), which all play significant roles for RNA structures and functions (Reyes et al., 2009; Vicens & Kieft, 2022).

Structure-based RNA design considers the inverse problem: Given a target structure, find an RNA primary sequence that folds into the desired structure. It is thus intricately tied to RNA folding. However, there is currently no structure-based RNA design algorithm available that can invert state-of-the-art deep learning-based secondary structure prediction algorithms, which could offer substantially improved designs, crucial for synthetic biology and the development of RNA-based therapeutics.

In this work, we propose RNAinformer, the first inverse RNA folding algorithm that is capable of designing RNAs while considering all kinds of base interactions. Inspired by the RNAformer (Franke et al., 2024), we show that a transformer architecture enhanced with axial attention can reliably design RNAs in different settings including RNA design with non-canonical interactions, pseudoknots, and base multiplets. Figure 1 shows example designs of the RNAinformer solving tasks that

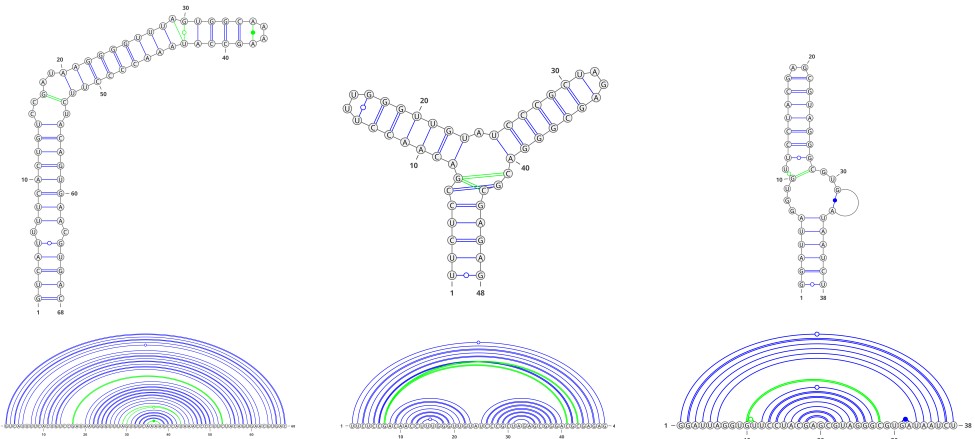

Figure 1: Example designs for experimentally validated RNA structures that include non-canonical base pairs, pseudoknots, and base multiplets. We show predictions of the RNAinformer that solve the respective structures.

contain non-canonical base pairs, pseudoknots, and base multiplets. We see our main contributions as follows:

- We propose RNAinformer, a novel generative transformer model for the inverse RNA folding problem. Using axial attention, our model is the first RNA design algorithm that can design RNAs from secondary structures with all types of base interactions (Section 3).
- We present a data pipeline for creating synthetic datasets for RNA design, with data splits based on RNA families (Section 4).
- We show that our model outperforms existing algorithms on nested and pseudoknot structures, while further being capable of designing sequences that form base multiplets (Section 5).

Our source code, data, and trained models are publicly available[1].

## 2 RELATED WORK

**Traditional Methods**   The problem of computational RNA design was first introduced as the inverse RNA folding problem by Hofacker et al. (1994). Since then, different methods were proposed for solving the problem using approaches like local search (Hofacker et al., 1994; Andronescu et al., 2004), constraint programming (Garcia-Martin et al., 2013; 2015; Minuesa et al., 2021), evolutionary methods (Esmaili-Taheri et al., 2014; Esmaili-Taheri & Ganjtabesh, 2015), or multifrontier search (Zhou et al., 2023). However, in contrast to our approach, these methods are limited to the design of nested structures, typically considering canonical base pairs only.

**Learning Based Approaches**   More recently, RNA design was also approached with learning based methods. One line of research use human priors to design RNAs based on player strategies obtained from the online gaming platform Eterna (Shi et al., 2018; Koodli et al., 2019). However, these models incorporate human strategies that might not be available for all designs and consider nested structures only. The other, more general approach seeks to learn RNA design purely from data. Eastman et al. (2018) propose to use reinforcement learning (RL) to adjust an initial input sequence by replacing nucleotides based on structural information. In contrast, Runge et al. (2019) and Riley et al. (2023) use a generative approach to the problem. Runge et al. (2019) employs a joint architecture and hyperparameter search approach (Bansal et al., 2022) via automated reinforcement learning (AutoRL) (Parker-Holder et al., 2022) to derive an RL system that is capable of generatively designing RNAs that fold into a desired target structure. Riley et al. (2023) uses a GAN (Goodfellow et al., 2020) approach specifically for the design of toehold switches (Green et al., 2014). However,

---

[1]An anonymized repository is available at `https://anonymous.4open.science/r/RNA-design-7204/`

all learning-based approaches so far consider RNA design for nested structures only, ignoring pseudoknots and base multiplets, while often being limited to the design of canonical base interactions.

**Pseudoknotted Structures**  Pseudoknots are an important type of base pairs that influence the function of an RNA (Staple & Butcher, 2005). Therefore, some approaches tried to design RNAs from pseudoknotted structures (Taneda, 2012; Kleinkauf et al., 2015; Merleau & Smerlak, 2022). However, these algorithms work on a string notation in dot-bracket format (Hofacker et al., 1994), and thus, they cannot express base multiplets.

Overall none of the existing algorithms can design RNAs including non-canonical base pairs, pseudoknots, and base multiplets.

**RNA Design from 3D Structures**  Besides the described approaches to design RNA based on secondary structure information, recently different methods also tackled the design of RNA sequences based on 3D structure information. The current state-of-the-art physics-based toolkit for biomolecular modeling and design is Rosetta (Leman et al., 2020). However, recently deep learning approaches challenged Rosetta's performance. Joshi & Liò (2024) developed gRNAde, a geometric deep learning-based RNA design pipeline that can be conditioned on RNA 3D backbone structures. Similarly, RDesign (Tan et al., 2024), a hierarchical framework that leverages a contrastive learning approach and incorporates secondary structure information, and RiboDiffusion (Huang et al., 2024), a diffusion model based on a graph neural network (GNN) (Zhou et al., 2020) structure- and a transformer-based (Vaswani et al., 2017) sequence module, showed remarkable results.

However, in contrast to RNAinformer, these methods leverage additional 3D information for their designs and, therefore, tackle RNA Design from a different perspective. Often, 3D information of RNA structures is not available and RNA 3D structure prediction is still challenging (Das et al., 2023). Therefore, strong secondary structure based RNA design approaches are highly thought after, also, but not limited to, to achieve better 3D predictions.

## 3  THE RNAINFORMER

RNA secondary structures can be represented in multiple ways, including the common dot-bracket string notation (Hofacker et al., 1994) or adjacency matrices. We show different representations in Figure 8. One advantage of an adjacency matrix representation is that it can model all types of base interactions, especially if a nucleotide interacts with more than one other, a situation prevalent for most experimentally solved structures (Singh et al., 2019). In the following, we detail our generative approach to designing RNAs from secondary structures using matrix representations.

**Model**  Our model is a modified auto-regressive encoder-decoder transformer model (Vaswani et al., 2017) with a next token prediction objective. The encoder embeds the structure information, while the decoder auto-regressively generates RNA nucleotide sequences by sampling from the softmax distribution (see Figure 5 in Appendix A). For RNAinformer we use axial attention in the first encoder block to process the adjacency matrix input similar to the RNAformer (Franke et al., 2024) (Figure 6 in Appendix A). To reduce the memory footprint of the 2D latent operations, we use flash-attention-2 (Dao, 2023) in the axial attention modules. For computational efficiency, we use pooling to reduce the 2D latent representation to a 1D vector that is then passed through the encoder and the decoder to generate candidate sequences. During constrained generation, we pass an additional input of the masked RNA sequence to the encoder. The masked sequence is embedded into a 2D representation and concatenated to the structure embedding. Similarly, for property conditioning, we embed the target GC-Content using a linear layer and add it to the structure embedding (see Figure 7 in Appendix A). For more details about the RNAinformer architecture and the formulation of the loss, please see Appendix A and B.

## 4  A HOMOLOGY AWARE SYNTHETIC DATA PIPELINE FOR RNA DESIGN

While secondary structure information obtained from experimentally validated RNA 3D structure data is considered the gold standard, this data is scarce; only roughly 3% of all available 3D struc-

ture data contains RNAs (Schneider et al., 2023). Therefore, most of the available training data is typically derived from comparative sequence analysis (Choudhary et al., 2017) and, thus, is less reliable. Further, the diversity within existing training sets is limited to only a few RNA families, making it challenging for a folding algorithm to generalize to less represented RNA types (Flamm et al., 2021). This recently raised skepticism in the RNA community that learning based models might not be able to generalize to unseen families (Flamm et al., 2021; Szikszai et al., 2022), which could be a serious concern when using a folding algorithm to validate a given design. To evade this problem, we train the RNAinformer exclusively on synthetic data. This allows us to generate large amounts of training data while enabling us to create a family-based split of the data to avoid learning homologies; a known problem in the RNA folding community (Rivas et al., 2012). In the following, we detail our approach to generate clean family-based synthetic train-/test splits for training the RNAinformer in different settings.

**Initial Training Data Pool**   We generate an initial training data pool using the families of the Rfam database (Version 14.10) (Kalvari et al., 2020). We select all families with covariance models having a maximum CLEN (the number of columns from a sequence alignment defined as consensus (match) columns) of 500 and sample 1,000 sequences for each family from the covariance models using Infernal (Nawrocki & Eddy, 2013). However, while our initial length cutoff is set to 500, roughly 80% of the samples had a max length below 200 nucleotides. Since we use the provided test sets from Singh et al. (2021), which all have a maximum length below 200 nucleotides (see below), for our evaluations on known RNAs obtained from PDB, we decide to use a length cutoff at 200 nucleotides to decrease computational costs. The sampled sequences are then annotated using the Rfam covariance models and Infernal. Sequences that hit multiple families, families other than the family they were sampled from, or did not hit any of the families were removed. To reduce intra-clan and intra-family sequence similarity we use CD-HIT (Fu et al., 2012) with a 0.8 threshold to cluster the sequences within a clan and if there is no clan information then within the family. Families or clans with less than 50 clusters are removed and we keep a maximum of 300 representatives of the clusters for each family/clan.

**Data Splits**   We split the data into training, validation, and test sets based on the clan information. All families without a clan annotation are put into the training set. We randomly sample 30 and 25 clans for the test and validation sets. For each test clan, we sample 100 sequences to form a test set and 50 sequences from each validation clan to form a validation set. The samples from all other clans are used for training. We then apply CD-HIT with a similarity cutoff of 80% to remove sequence similarity between the training, validation, and test sets, followed by a BLAST-search (Altschul et al., 1997) to further remove training and validation samples that are hit by BLAST for any of the test samples at a high e-value of 10 similar to Singh et al. (2021) but for all test data.

**Obtaining Structure Information**   We fold all sequences using different folding algorithms to create three datasets with different structural complexity:

1. **SynNested**: Folded using RNAfold (Lorenz et al., 2011) containing only nested structures.
2. **SynPseudoknot**: Folded using HotKnots2.0 (Andronescu et al., 2010) containing both pseudoknotted and nested structures.
3. **SynMultiplet**: Folded using RNAformer (Franke et al., 2024) containing structures with all base interactions, including base multiplets and non-canonical base pairs.

Further filtering is done to remove structures with no base pairs and structure duplicates.

**Experimental Structures from PDB**   To evaluate RNAinformer on known RNAs, we use additional test sets, TS1, TS2, TS3, and TS_hard from Singh et al. (2021), derived from experimental structures of the Protein Data Bank (PDB) (Berman et al., 2000). All the test sets contain structures with non-canonical base pairs, pseudoknots and base multiplets. To ensure non-homologous data, we apply an additional homology pipeline to remove homologous RNAs from SynMultiplet that share any sequence or structure similarity to any test sample. Specifically, we build covariance models from multiple sequence alignments for every test RNA employing LocARNA-P (Will et al., 2012) and remove any sequences from the training and validation sets that have a hit with any of the resulting covariance models using Infernal as previously described (Runge et al., 2024a). This ensures that there is no data homology between the training and test sets based on structure and sequence similarity.

All datasets used for our experiments are detailed in Appendix C.

## 5 EXPERIMENTS

We evaluate the RNAinformer on three RNA design paradigms, inverse RNA folding, constrained design, and RNA design with desired properties. We show that the RNAinformer can approach inverse RNA folding and RNA design with desired properties for tasks with increasing structural diversity by first evaluating RNA design for nested structures in Section 5.1, before tackling RNA design for pseudoknotted structures in Section 5.2. We then demonstrate RNAinformer's capability to conditionally generate RNA sequences for the real-world task of designing theophylline riboswitches following Runge et al. (2024a) in Section 5.3. We conclude with an assessment of RNAinformer's ability to design RNAs from secondary structures that contain all kinds of base interaction in Section 5.4. Here, we also compare our strategy to train on synthetic data with a more commonly used fine-tuning strategy using two versions of the RNAinformer.

During evaluation, we generate 20 candidate sequences with the RNAinformer for each task except for the PDB structures where we instead generate 100 candidate sequences. The first sequence is generated using a greedy strategy and the rest are generated using multinomial sampling. We set the threshold for satisfying the property constraint as $\epsilon = 0.01$.

**Metrics** The ultimate goal of structure-based RNA design is to generate sequences that fold back into the target structure. Following the common convention in the field of RNA design, we report the number of solved tasks for a given benchmark dataset. However, we provide a more comprehensive analysis of all experiments with different performance measures, described in detail in Appendix D.2.

**Training Details** We train our model with 6 encoder blocks and 6 decoder blocks with an embedding dimension of 256. The model is trained using cosine annealing learning rate schedule with warm-up and AdamW (Loshchilov & Hutter, 2019). We train separate models for each training dataset and for each training dataset we also train a separate GC-content conditioned model. The constrained design models were trained with a maximum length of 100 while the rest were trained with length 200. The longer models were trained across 2 A40 GPUs with an effective batch size of 128 for 50,000 steps having a runtime of ~18 hours. The GC-content of the original sequences is used as the target GC-content. The hyperparameters used for training our model are described in Table 2 in Appendix B.

### 5.1 RNA DESIGN FOR NESTED STRUCTURES

We first evaluate the RNAinformer's ability to design RNA sequences for nested structures with only canonical base pair interactions on the SynNested test set (see Table 3 in Appendix C). Additionally, we also evaluate its ability to design sequences with desired GC-content. We compare the performance of the RNAinformer for inverse folding against one of the currently best-performing set of algorithms, LEARNA, Meta-LEARNA, Meta-LEARNA-Adapt (Runge et al., 2019), libLEARNA (Runge et al., 2024b) and SAMFEO (Zhou et al., 2023). For design with desired GC-content we compare it against libLEARNA. The LEARNA suite algorithms and libLEARNA were run with a timeout of 30 seconds per sample and SAMFEO was run for 1,000 iterations for each task. We then select the best 20 candidates for evaluation. Corresponding to the data generation, the designed candidates are folded using RNAfold Lorenz et al. (2011) for evaluation; note that all competitors also used RNAfold for training and/or evaluation in their respective original publications.

### RESULTS

**Inverse Folding** From Table 1 we observe that RNAinformer outperforms most of its competitors except SAMFEO, solving 91.8% of the tasks. Furthermore, RNAinformer generates multiple, highly diverse solutions for each task, indicated by a high diversity score of 0.699 as shown in Table 10 in Appendix E.1. Remarkably, this performance is achieved by sampling only 20 sequences from the RNAinformer without any post-processing strategies as e.g. implemented in the local search strategy of the LEARNA suite of algorithms. However, while SAMFEO is capable of solving more

Table 1: Performance on the nested and pseudoknotted structures of the SynNested and SynPseudo-knot datasets, respectively, for Inverse Folding (IF) and Desired GC-content design (GC). We report the % tasks solved in 20 designed sequences.

| Model | SynNested | | SynPseudoknot | |
|---|---|---|---|---|
| | Solved (IF) [%] | Solved (GC) [%] | Solved PK (IF) [%] | Solved PK (GC) [%] |
| RNAinformer | 91.8 | **69.6** | **68.5** | **33.7** |
| LEARNA | 63.9 | ✗ | ✗ | ✗ |
| Meta-LEARNA | 36.8 | ✗ | ✗ | ✗ |
| Meta-LEARNA-Adapt | 37.2 | ✗ | ✗ | ✗ |
| libLEARNA | 77.2 | 59.0 | ✗ | ✗ |
| SAMFEO | **99.6** | ✗ | ✗ | ✗ |
| antaRNA | ✗ | ✗ | 15.6 | 1.2 |

tasks, it generates solutions with much lower diversity (0.106) compared to the RNAinformer. One reason is that SAMFEO uses an initialization strategy that itself could already solve 78% of the tasks, leveraging biases in the internal scoring functions of RNAfold by placing low energy GC-pairs at paired positions and single A nucleotides at unpaired positions that cannot pair with G or C in the limited model of RNAfold. Despite increasing the ability to solve the design tasks, this approach has the disadvantage that the resulting candidates rarely contain U nucleotides, typically resulting in high GC nucleotide ratios (GC-content), which can drastically impact the function of the resulting RNAs (Isaacs et al., 2006). In contrast, the designs of the RNAinformer do not show similar bias as indicated by the high sequence diversity.

**Desired GC-Content Design**  The GC-content conditioned RNAinformer model solves ∼10% more tasks than libLEARNA, the only competitor capable of also generating RNAs with desired GC-contents, as shown in Table 1. Further, our results in Table 11 in Appendix E.1 demonstrate that even with the GC-content constraints, RNAinformer can still generate multiple highly diverse solutions for each task. We also note that the average GC-content error of the candidate sequences generated by RNAinformer is very low (0.01), indicating that the model actively generates sequences with GC-content close to the desired target value.

## 5.2  RNA Design with Pseudoknots

In this section, we assess the performance of RNAinformer when designing RNAs for pseudoknotted input structures. Pseudoknots are tertiary interactions that typically connect local geometries of the RNA secondary structure by establishing long-range interactions between nucleotides. We compare the RNAinformer against antaRNA (Kleinkauf et al., 2015) with HotKnots2.0 (Andronescu et al., 2010) as the folding algorithm; one out of three folding engines available for antaRNA which was also used for data generation. Again, we provide results for both inverse-folding and for the design with desired GC-content. We evaluate the RNAinformer on both the pseudoknotted structures(pK) and nested structures(pK-free) of the SynPseudoknot Dataset (see Table 4 in Appendix C). However, due to the long runtime of antaRNA and its internal ant-colony optimization strategy, we only evaluate one design candidate, supposed to solve the task, and limit the comparison to the pseudoknotted structures (pK). However, there are many more intermediate sequences designed by antaRNA before outputting the final design. To make a fair comparison, we additionally evaluate the first design of RNAinformer for completeness.

### Results

**Inverse Folding**  RNAinformer significantly outperforms antaRNA, solving ∼50% more pseudo-knot tasks, as shown in Table 1 (right). Remarkably, RNAinformer also achieves high performance on the nested structures solving more than 90% of the tasks as shown in Table 12 in Appendix E.2. Generally, we observe that RNAinformer is able to generate multiple solutions with high diversity for both the nested and the pseudoknotted structures. The high F1 and MCC scores of RNAinformer further indicate that the designs are close to a solution even for the tasks that could not be solved

with 20 candidates (see Table 12 in Appendix E.2). Notably, the RNAinformer also solves 39.1% of the tasks with the first sequence generated, still outperforming antaRNA by solving twice the number of tasks (see Table 13 in Appendix E.2).

**Desired GC-Content Design**   Similar to the unconditional generation, the RNAinformer also outperforms antaRNA for the conditional design of RNAs with desired GC-contents by a large margin as shown in Table 1. The RNAinformer roughly solves one third of the pseudoknotted structures (33.7%) compared to only 1.2% solved tasks by antaRNA. Although the number of solutions generated by the conditioned RNAinformer model is less compared to the unconditioned model, the diversity of the solutions is maintained as shown in Table 13 in Appendix E.2. Moreover, the RNAinformer is capable of solving almost two-thirds of the nested structures (65.8%) while again generating sequences with low GC-content error, indicating closeness to the target value (see Table 13 in Appendix E.2). Again we evaluate the first prediction of RNAinformer and observe that it solves nearly 15 times the number of tasks compared to antaRNA (solving 17.7% of the tasks with a single shot; see Table 13 in Appendix E.2).

## 5.3 AUTOMATED DESIGN OF THEOPHYLLINE RIBOSWITCHES

We evaluate the RNAinformer's ability to do constrained design by tackling the design of synthetic theophylline riboswitches. We use the design space formulation from Runge et al. (2024b), which was created by combining the shared sequence and structure motifs of the proposed constructs by Wachsmuth et al. (2012), defined as,

$$\hat{\omega} = \ldots\ldots\ldots\text{???}((((\,(\ldots\ldots)\,))))\,\ldots\text{???}(((((((((\,(\hat{?}\text{??}\ldots\hat{?})\,)))))))))\,)\hat{?}\text{?}\ldots\ldots$$

$$\hat{\phi} = \text{AAGUGAUACCAGCAUCGUCUUGAUGCCCUUGGCAGCACUUCA}\hat{?}\text{??????}\hat{?}\text{UGAAGUGCUG}\hat{?}\text{UUUUUUUU}$$

$$(1)$$

where ? represent masked out positions and $\hat{?}$ represent positions for extensions. The different sections of the construct are highlighted, (i) Aptamer (Red), (ii) Spacer (Green), (iii) the Complementary Sequence (Blue), and (iv) the 8-U Stretch (Black). We use the above formulation to sample tasks for our evaluation. Since we do not have any ground truth sequences to get target GC-contents we test RNAinformer's ability for conditional generation on a range of GC-content values for each task. We compare our models against libLEARNA for both inverse folding and design with desired GC-content. For each task, we again generate 20 candidates and use RNAfold to fold them, in line with the original procedure described at Runge et al. (2024b).

**Training Data**   For training, we use the Training Short and Validation datasets provided by Runge et al. (2024b) (see Table 7 in Appendix C). The datasets were generated by sampling from the Rfam database version 14.1 and folding all the sequences with RNAfold. The structures and sequences were then randomly masked to create the final datasets for constrained design. During training, target GC-content values were obtained from the unmasked sequences.

**Riboswitch Tasks**   We generate an exhaustive set of riboswitch design tasks for evaluation using the design formulation 1. We sample masked sequences for each of the extension positions while considering the length constraints for each part. We filter the tasks for the seven GC-content targets, listed in Table 8 in Appendix C, based on the possible range of GC-contents for each task as calculated from their length and masked sequence. Few of the generated tasks have no valid sequences possible when evaluating using RNAfold. As it is not feasible to determine all the un-designable tasks we do not filter them out.

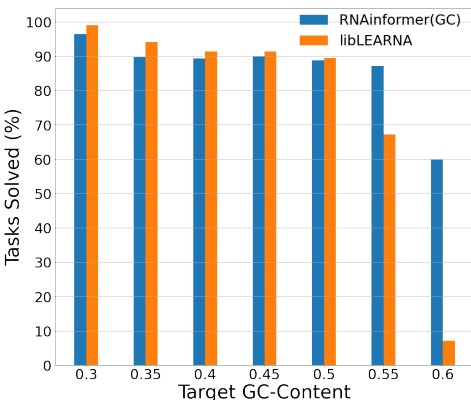

Figure 2: Comparison between RNAinformer and libLEARNA for Riboswitch design with desired GC-content.

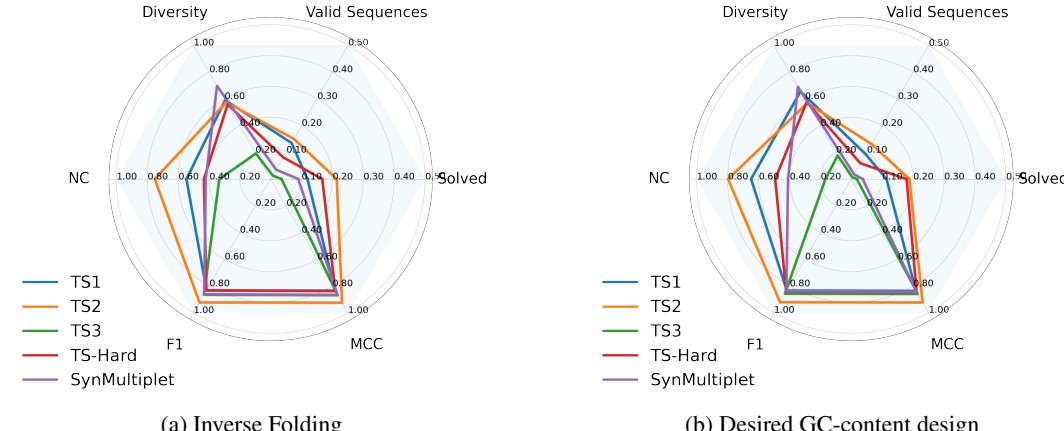

(a) Inverse Folding    (b) Desired GC-content design

Figure 3: Results for RNA Design with all kinds of base pair interactions.

## RESULTS

**Inverse Folding**    Both RNAinformer and libLEARNA are capable of solving almost all of the riboswitch tasks (solving >90% of the tasks) as shown in 14 in Appendix E.3. However, the RNAinformer slightly outperforms libLEARNA. Furthermore, the RNAinformer generates more solutions (valid sequences) compared to libLEARNA. Due to the fixed sequence constraints, the observed diversity of the solution sequences is rather low.

**Desired GC-Content Design**    The results for the desired GC-content design are shown in Figure 2. We observe that libLEARNA outperforms RNAinformer for the smaller GC-content targets (0.3,0.35), whereas RNAinformer outperforms libLEARNA for the larger target values (0.55,0.6) and the performance is almost identical for the mid-range targets. However, the performance of the RNAinformer is generally significantly more consistent across all the target GC-content values. Additional results are shown in Table 15 in Appendix E.3.

## 5.4    RNA DESIGN WITH ALL KINDS OF BASE INTERACTIONS

In this section, we evaluate RNAinformer for designing RNAs from structure data that contains all kinds of base pairs including pseudoknots and base multiplets on experimentally validated structures from the PDB (Berman et al., 2000) and the SynMultiplet Dataset (see Table 5 in Appendix C). To account for the difficulty of the task, we design 100 candidate sequences instead of only 20 sequences. The RNAinformer is the only method capable of tackling the task of designing RNAs for structures that contain base multiplets; consequently, we cannot compare our designs with other methods from the field. Instead, we compare against a simple baseline that uniformly samples RNA sequences for both inverse folding and desired GC-content design and a GNN baseline where we use the implementation of structTransformer provided by Ingraham et al. (2019) and run it with the same batch size and steps as the RNAinformer. According to the data generation, the designed candidate sequences are folded using RNAformer Franke et al. (2024) for evaluation.

## RESULTS

**Inverse Folding**    RNAinformer significantly outperforms the randomly designed sequences and the GNN baseline as shown in Table 16. From Figure 3a we observe that RNAinformer further is the first method capable of solving experimentally determined structures across the PDB test sets including structures with base multiplets. However, the overall solved rate is rather low, indicating that designing sequences for structures with all kinds of base pairs seems to be much more challenging than for nested structures or structures with pseudoknots only. Examples of solved experimental structures with base multiplets are shown in Figure 1. Despite relatively low rates of solved tasks, the RNAinformer still achieves high F1 and MCC scores, indicating that the designed candidates have high structural accuracy. Notably, the RNAinformer achieves similar performance on the Syn-

Multiplet Test set. In addition, the RNAinformer is able to generate multiple solutions with high diversity for all the test sets except TS3 and can generate solutions with non-canonical base pairs, which are typically ignored by other design algorithms.

**Desired GC-Content Design** As shown in Figure 3b, the GC-conditioned RNAinformer can solve tasks across the PDB test sets including structures with base multiplets, even with the additional GC-content constraint. While the number of solutions generated drops significantly, the diversity of the generated sequences is maintained. Similar to the unconditioned model, we observe high F1 and MCC scores indicating structural similarity to the ground truth, and that the RNAinformer generates solutions with non-canonical base pairs and candidate sequences with a low GC-content error across the test sets.

Detailed results are shown Table 17 and Table 18 in Appendix E.4.

**Synthetic data vs Real-World data** To validate our strategy to only use synthetic data during training of the RNAinformer, we additionally train a model on known RNA data using the inter-family dataset from Runge et al. (2024a). The training data was collected from multiple public sources: bpRNA-1m (Danaee et al., 2018), ArchiveII (Sloma & Mathews, 2016) and RNAStrAlign (Tan et al., 2017) from Chen et al. (2020), RNA-Strand (Andronescu et al., 2008) and PDB (Berman et al., 2000). Homologies between the training, validation, and test sequences were removed by filtering using CD-Hit (Fu et al., 2012) and BLAST-Search (Altschul et al., 1997). An additional homology reduction based on structure similarity was applied using covariance models of the PDB test sets (TS1, TS2, TS3 and TS-Hard). We train an RNAinformer model (NAT + FT) on it and further fine-tuned it on the PDB training samples. We compare it against an RNAin-

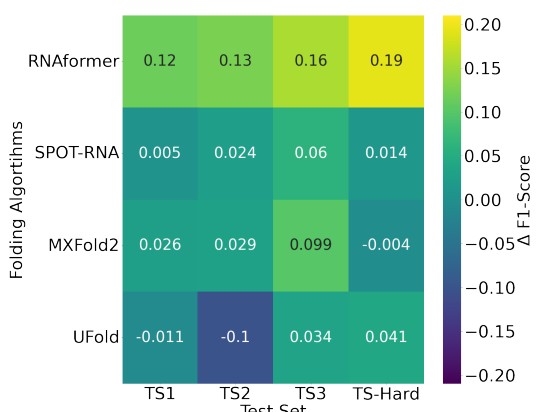

Figure 4: Difference in F1-Scores between the folded structures for the designed sequences and the PDB test set sequences.

former model (Syn) pre-trained on the SynMultiplet dataset and a second model that was also pre-trained on synthetic data but finetuned on the PDB samples (Syn + FT) similar to the NAT + FT model.

The results are shown in Table 19 in Appendix E.4. Surprisingly, the RNAinformer model pre-trained on synthetic data performs significantly better than the model that was pre-trained and fine-tuned on known RNAs, having nearly double the F1 score across the test sets. However, the additional finetuning appears beneficial, as indicated by slightly higher scores of the Syn + FT model. To further assess this, we evaluated the Syn and Syn + FT models on RNA only samples from the Critical Assessment of Structure Prediction 15 (CASP15) competition. As before, we use the RNAformer for secondary structure predictions but additionally employ AlphaFold 3 (Abramson et al., 2024) for 3D structure prediction of the generated sequences. The results are shown in Tables 20 and 24 for 2D and 3D predictions, respectively. In contrast to the results for the PDB test set, we observe that the finetuned model achieves slightly worse performance than the model trained on synthetic data only for both folding engines, RNAinformer and AlphaFold 3. We conclude that training on synthetic data appears beneficial for RNAinformer compared to training on known RNAs only, and finetuning on experimentally validated structures can lead to slightly better performance in some cases.

**Improved Foldability of Designed Sequences** The RNAinformer's designed sequences on average achieve higher structure F1 scores across the PDB test sets compared to RNAformer's original predictions on the PDB test set sequences. We take this as an indicator that the RNAinformer learns to design sequences that are better foldable by the RNAformer for the experimental test structures. However, to ensure that these results are not artifacts resulting from overfitting the RNAformer's

distribution, we folded the best-designed sequences by the RNAinformer for each of the PDB test structures using other folding algorithms (SPOT-RNA (Singh et al., 2019), MXFold2 (Sato et al., 2021) and UFold (Fu et al., 2022)) and compared the F1 scores against the folding algorithms' predictions on the PDB test set sequences. We observe that designed sequences have better or similar F1 scores for almost all the folding algorithms, showing consensus agreement for the designed sequences (see Figure 4). The F1 scores of the designed sequences and the folding algorithms evaluations on the PDB test sets are reported in Table 21 in Appendix E.4. We again also folded the designed sequences with AlphaFold 3 and analyzed the results regarding TM score and RMSD. The results are shown in Table 22. We find that RNAinformer cannot improve the TM scores compared to TM scores achieved when folding the ground truth sequence with AlphaFold 3. However, AlphaFold 3 is known to struggle with predictions for so-called orphan RNAs (Bernard et al., 2024) – sequences where there exist no homologs in the database – due to its dependence on multiple sequence alignments (MSAs). Therefore, we analyzed the AlphaFold 3 predictions in more detail and find that for all the designed sequences AlphaFold 3 was not able to find an MSA during the search while roughly 59% of the PDB samples had multiple homologs. We show the difference in TM-Score when splitting the data into orphan and non-orphan RNAs in Table 23. We observe that the designed sequences of the RNAinformer slightly improve the TM score for three out of four datasets for orphan RNAs. However, when there is MSA available for the ground truth but not for the designed sequences, the TM score drops drastically for the designed candidates.

## 6 Conclusion, Limitations & Future Work

In this work, we propose RNAinformer, the first RNA design algorithm capable of designing RNA sequences from secondary structures that contain all kinds of base interactions, including non-canonical base pairs, pseudoknots, and base multiplets. Using axial-attention, the RNAinformer leverages a 2D latent representation to process adjacency matrix representations of RNA secondary structures to achieve state-of-the-art results in structure based RNA design. We demonstrate the strong performance of RNAinformer on tasks with nested structures only, tasks that contain pseudoknots, as well as on experimentally derived structures with all kinds of base interactions. We observe high diversity across all designs and tasks and improved foldability of the designed sequences compared to their known counterparts.

**Limitations**   While showing overall strong performance, there is still room for improvement, particularly for the design for known RNA structures. Further, while we reduce computational complexity using a pooling operation to map the 2D latent representation to a 1D vector, training the RNAinformer is memory intensive. As a result, we only train the RNAinformer with a sequence length cutoff at 200 nucleotides. While this is sufficient for current benchmarks, a higher length cutoff would further increase the usability of our approach.

**Future Work**   We think that RNAinformer is a useful basis for future approaches to RNA design and expect it to be of great value for the RNA design community. Future work could e.g. focus on improving the memory footprint of the RNAinformer.

## REPRODUCIBILITY STATEMENT

To ensure the reproducibility of our results, we have made our source code, model checkpoints, and datasets publicly available in the anonymous repository `https://anonymous.4open.science/r/RNA-design-7204/`. The repository contains detailed instructions for setting up the environment, including specific Python package versions (see `environment.yml`). Model checkpoints and predictions for all experiments are provided in the `runs.ta.xz` file. Our datasets are provided in the `data.tar.xz`. Links to download both files are in the repository. We provide scripts for evaluating trained models (`eval.py`) and for reproducing our training procedures (`train.py` with configs for all trained models also provided. Scripts for running inference on the test sets is also provided (`inference.py`). Hardware requirements (GPU specifications) for both training and inference are clearly stated. Evaluation on provided predictions can be done without gpus.

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

# A    MODEL DETAILS

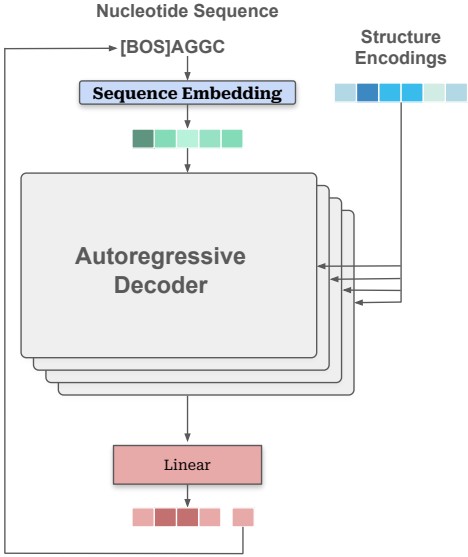

Figure 5: Overview of nucleotide sequence generation.

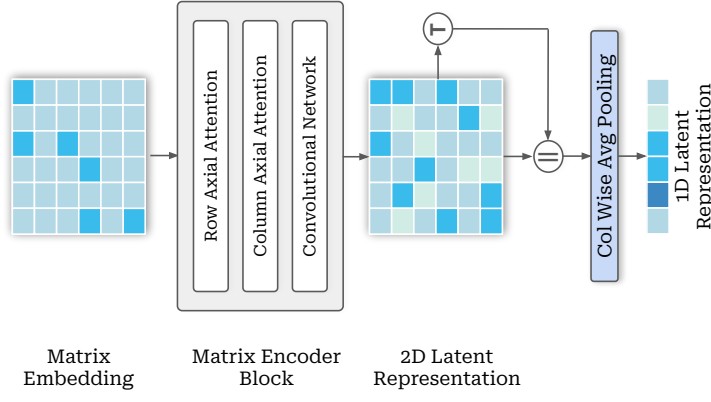

Figure 6: Overview of matrix input processing in RNAinformer.

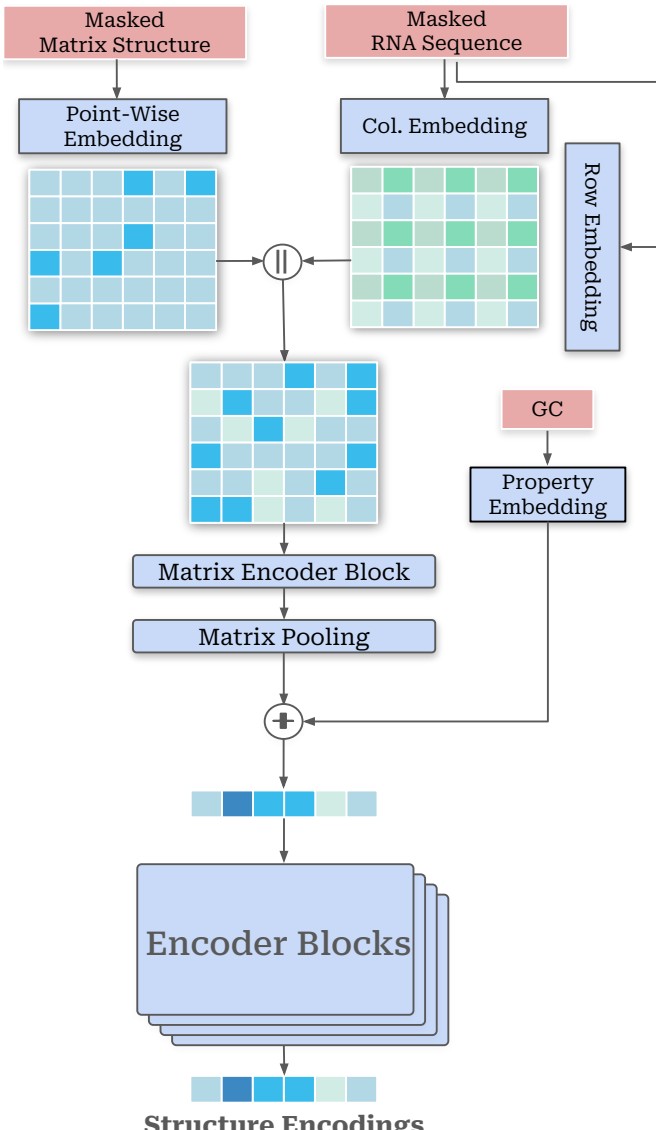

Figure 7: Overview of RNAinformer encoder for constrained design and GC-Content conditioning with an adjacency matrix structure representation.

## B    TRAINING DETAILS

**Loss**    The problem of RNA design is often addressed by defining a structural loss function $L_\omega = d(\omega, \mathcal{F}(\phi))$ that quantifies the difference between the target structure $\omega$ and the folding, $\mathcal{F}(\cdot)$, of the designed candidate sequence $\phi$ (Runge et al., 2019). However, the folding process is generally not differentiable making it difficult to use the structural loss for training in deep learning based approaches. Instead, we cast the problem as a conditional language modeling problem and train a conditional transformer language model on RNA sequences.

The inverse folding problem can then be formulated as conditioning RNA sequences on the target structures. The conditional probability for an RNA sequence $\phi$ conditioned on a target structure $\omega$ is,

$$p(\phi) = p(\phi \mid \omega) \qquad . \qquad (2)$$

To design RNA sequences with a certain set of desired properties $C$ we extend the inverse folding formulation and condition the sequence on both the target structure $\omega$ and the desired properties $C$. Equation 2 is extended to,

$$p(\phi) = p(\phi \,|\, \omega, C) \qquad . \tag{3}$$

Similar to the task of scaffold-based generation for small molecules (Bagal et al., 2021) but for both the RNA sequence and the structure, we can further extend the above formulation for constrained design to include constraints on the target structure as well as the designed sequence at certain positions. By imposing these constraints on the sequence $\phi$ and the structure $\omega$ we get the masked sequence $\hat{\phi}$ and the masked structure $\hat{\omega}$, respectively, to condition the sequence generation on. Equation 3 then becomes,

$$p(\phi) = p(\phi \,|\, \hat{\omega}, \hat{\phi}, C) \qquad . \tag{4}$$

Using auto-regressive modeling allows us to factorize the probability of the whole sequence $p(\phi)$ into,

$$p(\phi) = \prod_{i=1}^{l} p(\phi_i \,|\, \phi_{<i}, \hat{\omega}, \hat{\phi}, C) \qquad , \tag{5}$$

where $l$ is the length of the sequence $\phi$.

This decomposes the design problem into a next token prediction problem. Now we can train a model with parameters $\theta$ over a dataset $\mathcal{D} = \{(\phi, \hat{\omega}, \hat{\phi}, C)\}^n$, where $n = |\mathcal{D}|$ using the loss,

$$\mathcal{L}_{\mathcal{D}} = \frac{1}{n} \sum_{k=1}^{n} \frac{1}{l^k} \sum_{i=1}^{l^k} l_{CE}(\phi_i^k, \theta(\phi_{<i}^k, \hat{\omega}^k, \hat{\phi}^k, C^k)) \qquad , \tag{6}$$

where $L_{CE}(\psi_i, \phi_i)$ is the cross entropy loss between the target sequence and the designed sequence at position $i$.

Table 2: Hyperparmeters for RNAinformer training.

| Group | Parameter | Value |
|---|---|---|
| Trainer | Batch Size | 128 |
| | Training Steps | 50,000 |
| Optimizer | LR | 0.0005 |
| | Weight Decay | 0.1 |
| | Betas | 0.9,0.98 |
| LR Schedule | Schedule | Cosine Annealing |
| | LR Decay Factor | 0.1 |
| | Warmup Steps | 1,000 |
| Model | Model dim | 256 |
| | Layers | 6 |
| | Num Head | 4 |
| | FeedForward factor | 4 |
| | FeedForward kernel | 3 |
| | Dropout | 0.1 |

## C  DATASETS

Table 3: Overview of the SynNested dataset.

| Set | #Samples | Avg Length | Pseudoknots | Multiplets |
|---|---|---|---|---|
| Train | 444766 | 100 | 0(0.00%) | 0(0.00%) |
| Valid | 1108 | 100 | 0(0.00%) | 0(0.00%) |
| Test | 2722 | 96 | 0(0.00%) | 0(0.00%) |

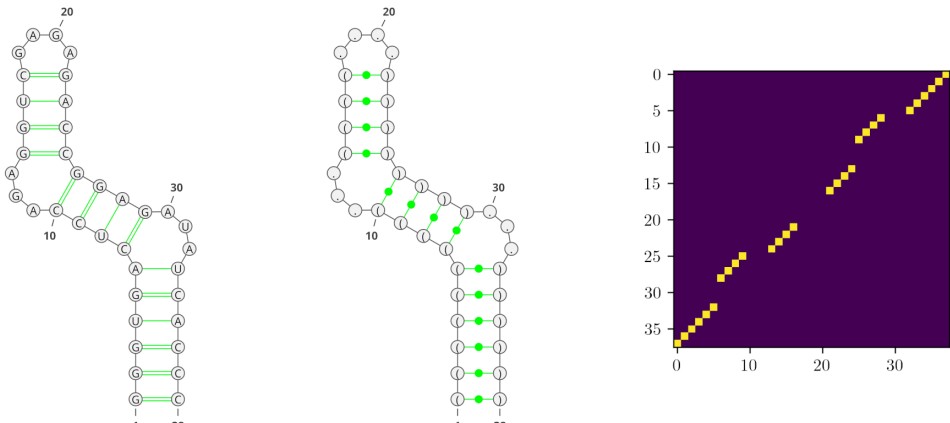

Figure 8: Representations of RNA secondary structures. (Left) Common graph representation of the RNA. (Middle) Dot-bracket notation in the graph structure. A pair of nucleotides is indicated by a pair of matching brackets, unpaired nucleotides are indicated by a dot. (Right) Matrix representation of the RNA. The matrix is a binary $L \times L$ square matrix, where $L$ is the sequence length of the RNA. Pairing nucleotides are shown in yellow.

Table 4: Overview of the SynPseudoknot dataset.

| Set | #Samples | Avg Length | Pseudoknots | Multiplets |
|---|---|---|---|---|
| Train | 444768 | 100 | 19.82% | 0.00% |
| Valid | 1108 | 100 | 20.31% | 0.00% |
| Test | 2732 | 96 | 29.61% | 0.00% |

Table 5: Overview of the SynMultiplet dataset.

| Set | #Samples | Avg Length | Pseudoknots | Multiplets | Non Canonical BP |
|---|---|---|---|---|---|
| Train | 441028 | 100 | 44.92% | 57.44% | 63.85% |
| Valid | 1101 | 101 | 46.91% | 61.25% | 68.14% |
| Test | 2721 | 95 | 45.61% | 57.00% | 62.27% |

Table 6: Overview of the PDB Test sets.

| Set | #Samples | Avg Length | Pseudoknots | Multiplets | Non-Canonical BP |
|---|---|---|---|---|---|
| TS1 | 67 | 74 | 83.58% | 79.10% | 92.54% |
| TS2 | 39 | 52 | 66.67% | 74.36% | 97.44% |
| TS3 | 19 | 79 | 94.74% | 94.74% | 94.74% |
| TS-Hard | 28 | 66 | 71.43% | 75.00% | 85.71% |

Table 7: Overview of the Rfam Constrained Design Dataset.

| Set | #Samples | Avg Length | Pseudoknots | Multiplets |
|---|---|---|---|---|
| Train | 51063 | 73 | 0.00% | 0.00% |
| Valid | 49 | 72 | 0.00% | 0.00% |

Table 8: Overview of the Riboswitch Tasks.

| | Inverse Folding | Target GC-Content | | | | | | |
|---|---|---|---|---|---|---|---|---|
| | | 0.30 | 0.35 | 0.40 | 0.45 | 0.50 | 0.55 | 0.60 |
| **#Tasks** | 1440 | 205 | 1275 | 1440 | 1440 | 1436 | 1220 | 364 |

Table 9: Overview of the Inter-family Dataset.

| Set | #Samples | Avg Length | Pseudoknots | Non-Canonical BP | Multiplets |
|---|---|---|---|---|---|
| Train | 19540 | 73 | 2047(10.47%) | 11114(56.70%) | 1330(6.80%) |
| Valid | 494 | 77 | 12(2.43%) | 287(57.86%) | 13(2.63%) |
| TS1 | 54 | 61 | 43(79.62%) | 49(90.74%) | 40(74.07%) |
| TS2 | 36 | 45 | 23(63.88%) | 35(97.22%) | 26(72.22%) |
| TS3 | 16 | 67 | 15(93.75%) | 15(93.75%) | 15(93.75%) |
| TS-Hard | 25 | 55 | 17(68.00%) | 21(84.0%) | 18(72.00%) |

# D  EVALUATION

## D.1  NOTATION

**Task**  We call designing sequences for a particular target structure a task. The task may also have additional constraints for design with desired properties and constrained design.

**Solved Task**  If a task has at least one designed sequence that folds back into the target structure and satisfies the other constraints of the task, then the task is considered to be solved.

**Candidate Sequence**  All the designed RNA sequences for a particular task are considered as its candidate sequences.

**Valid Sequence**  Candidate sequences that solve a task are considered valid sequences for the task.

**Valid Structure**  If a candidate sequence for a task folds back into the target structure or satisfies the constraints on the structure, then it has a valid structure.

## D.2  METRICS

All metrics for the RNAinformer are reported as the mean and standard deviation of three random seed runs.

**Solved**  As the main performance measure of the model we report the percent of solved tasks for a given benchmark dataset.

**Valid Sequences (Valid Seq.)**  We refer any candidate sequence that solves a task as a valid sequence. We measure the efficiency of the generative process by the number of valid sequences that are produced for each task.

$$ValidSequences = \frac{\#ValidSequences}{\#CandidateSequences} \tag{7}$$

**Diversity (Div.)**  To measure the diversity of the valid sequences generated for a target structure, we use the pairwise Hamming distance. For $N$ valid sequences of length $l$ the diversity is defined as,

$$Diversity = \frac{1}{N} \sum_i^N \sum_j^N \frac{1}{l} \sum_{k=1}^l H(S_{ik}, S_{jk}) \qquad , \tag{8}$$

where $H(S_{ik}, S_{jk})$ describes the positional Hamming distance:

$$H(S_{ik}, S_{jk}) = \begin{cases} 0 & \text{if } S_{ik} = S_{jk} \\ 1 & \text{else} \end{cases} . \tag{9}$$

**NC**   To measure the models ability to design with non-canonical base pair interactions we report the number of valid sequences containing non-canonical base pairs.

**GC-Content Error (GCE)**   For design with desired GC-content we report the property constraint violation of the candidate sequences with valid structures, given by:

$$GCE = abs(GC_{target} - GC_{Sequence}) \tag{10}$$

where $GC_{target}$ is the target GC-Content value and $GC_{Sequence}$ is the GC-Content of a candidate sequence.

**F1 Score**   The F1 Score is a commonly used performance measure to assess the quality of secondary structure prediction algorithms. It is based on the confusion matrix, which describes the number of true positives (TP), true negatives (TN), false positives (FP), and false negatives (FN) when comparing a predicted structure to the ground truth. The F1 score is the harmonic mean of precision and sensitivity, defined as:

$$F1 = \frac{2 \cdot TP}{(2 \cdot TP + FP + FN)} \tag{11}$$

**Matthews Correlation Coefficient (MCC)**   Compared to the F1 score that emphasizes on positives, the MCC is a more balanced measure (Chicco & Jurman, 2020). The MCC can be calculated as follows.

$$MCC = \frac{(TP \cdot TN) - (FP \cdot FN)}{\sqrt{(TP + FP) \cdot (TP + FN) \cdot (TN + FP) \cdot (TN + FN)}} \tag{12}$$

For each task in a test set, we take the maximum F1 and MCC scores achieved by a candidate sequence and report the average over these values across three random seeds.

**TM-score and RMSD**   We use US-align (Zhang et al., 2022) to get the TM-scores and RMSD value based on optimal structural alignment between the folded 3D structure of the designed sequences and the ground truth 3D structure from the PDB.

# E    ADDITIONAL RESULTS

## E.1    RNA DESIGN FOR NESTED STRUCTURES

Table 10: Results for the design of RNAs for nested structures of the SynNested Dataset.

| Model | Solved | Valid Seq. | Diversity | F1 | Seq. Rec. |
|---|---|---|---|---|---|
| RNAinformer | 0.918±0.020 | 0.299±0.005 | 0.699±0.001 | 0.992±0.005 | **0.394±0.002** |
| LEARNA | 0.639 | 0.372 | 0.457 | 0.989 | 0.371 |
| Meta-LEARNA | 0.368 | 0.244 | **0.747** | 0.975 | 0.366 |
| Meta-LEARNA-Adapt | 0.372 | 0.236 | 0.746 | 0.976 | 0.366 |
| libLEARNA | 0.772 | 0.740 | 0.733 | 0.987 | 0.370 |
| SAMFEO | **0.996** | **0.999** | 0.106 | **0.999** | 0.343 |

Table 11: Results for the design of RNAs for nested structures of the SynNested dataset with target GC content.

| Model | Solved | Valid Seq. | Diversity | F1 | Seq. Rec. | GC Error |
|---|---|---|---|---|---|---|
| RNAinformer(GC) | **0.696±0.025** | 0.247±0.004 | 0.701±0.003 | **0.993±0.002** | **0.395±0.002** | **0.010±0.001** |
| libLEARNA | 0.590 | **0.452** | **0.728** | 0.923 | 0.373 | 0.082 |

## E.2    RNA DESIGN WITH PSEUDOKNOTS

Table 12: Results for the design of RNAs including pseudoknots on the SynPseudoknot dataset.

| Model | Test Set | Solved | Valid Seq. | Diversity | F1 | Seq. Rec. |
|---|---|---|---|---|---|---|
| RNAinformer | pK-free | 0.934±0.016 | 0.213±0.018 | 0.692±0.002 | 0.995±0.002 | 0.392±0.001 |
|  | pK | **0.685±0.064** | 0.140±0.028 | 0.692±0.004 | **0.947±0.009** | **0.405±0.001** |
| RNAinformer-1 | pK-free | 0.670±0.025 | - | - | 0.920±0.007 | - |
|  | pK | **0.391±0.045** | - | - | 0.734±0.025 | - |
| antaRNA | pK | 0.156 | - | - | 0.788 | 0.253 |

Table 13: Results for the design of RNAs including pseudoknots on the SynPseudoknot dataset with target GC content.

| Model | Test Set | Solved | Valid Seq. | Diversity | F1 | Seq. Rec. | GC-Error |
|---|---|---|---|---|---|---|---|
| RNAinformer(GC) | pK-free | 0.658±0.023 | 0.147±0.009 | 0.691±0.002 | 0.993±0.003 | 0.395±0.001 | 0.010±0.002 |
|  | pK | **0.337±0.021** | 0.105±0.003 | 0.694±0.002 | **0.937±0.015** | **0.405±0.001** | **0.013±0.001** |
| RNAinformer(GC)-1 | pK-free | 0.447±0.008 | - | - | 0.931±0.016 | - | 0.010±0.001 |
|  | pK | **0.177±0.016** | - | - | **0.757±0.045** | - | **0.010±0.001** |
| antaRNA | pK | 0.012 | - | - | 0.747 | 0.257 | 0.063 |

## E.3    AUTOMATED DESIGN OF THEOPHYLLINE RIBOSWITCHES.

Table 14: Results for the design of Riboswitches

| Model | Set | Solved | Valid Seq. | Diversity |
|---|---|---|---|---|
| RNAinformer | Riboswitch Tasks | **0.919±0.002** | **0.599±0.031** | 0.182±0.003 |
| libLEARNA | Riboswitch Tasks | 0.915 | 0.557 | **0.190** |

Table 15: Results for the design of Riboswitches with target GC-Content.

| Target | Model | Solved | Valid Seq. | Diversity |
|--------|-------|--------|-----------|-----------|
| 0.3 | RNAinformer(GC) | 0.964±0.020 | **0.403±0.043** | **0.126±0.001** |
| | libLEARNA | **0.990** | 0.264 | 0.125 |
| 0.35 | RNAinformer(GC) | 0.898±0.039 | **0.511±0.077** | 0.128±0.004 |
| | libLEARNA | **0.941** | 0.421 | **0.142** |
| 0.4 | RNAinformer(GC) | 0.893±0.015 | **0.486±0.048** | 0.149±0.002 |
| | libLEARNA | **0.913** | 0.398 | **0.177** |
| 0.45 | RNAinformer(GC) | 0.899±0.023 | **0.431±0.039** | 0.145±0.002 |
| | libLEARNA | **0.913** | 0.388 | **0.189** |
| 0.5 | RNAinformer(GC) | 0.888±0.036 | 0.399±0.039 | 0.143±0.002 |
| | libLEARNA | **0.894** | 0.309 | **0.174** |
| 0.55 | RNAinformer(GC) | **0.871±0.010** | **0.336±0.033** | 0.128±0.009 |
| | libLEARNA | 0.672 | 0.205 | **0.148** |
| 0.6 | RNAinformer(GC) | **0.600±0.083** | **0.193±0.014** | **0.128±0.010** |
| | libLEARNA | 0.071 | 0.050 | 0.000 |

## E.4 RNA DESIGN WITH ALL KINDS OF BASE INTERACTIONS

Table 16: Comparison with different baselines on the experimentally validated structures from PDB.

| Test Set | RNAinformer | | Random | | GNN | |
|----------|-------------|---------|--------|---------|-----|---------|
| | F1 | Seq. Rec | F1 | Seq. Rec | F1 | Seq. Rec |
| TS1 | **0.832±0.004** | **0.461±0.003** | 0.223±0.006 | 0.391±0.001 | 0.221±0.017 | 0.393±0.007 |
| TS2 | **0.923±0.004** | **0.479±0.007** | 0.349±0.014 | 0.416±0.007 | 0.315±0.04 | 0.405±0.004 |
| TS3 | **0.866±0.010** | **0.454±0.001** | 0.211±0.004 | 0.384±0.009 | 0.223±0.044 | 0.392±0.009 |
| TS-Hard | **0.834±0.007** | **0.465±0.006** | 0.274±0.023 | 0.401±0.003 | 0.283±0.016 | 0.407±0.012 |
| SynMulitplet | **0.862±0.002** | **0.432±0.001** | 0.205±0.002 | 0.371±0.001 | – | – |

Table 17: Results for RNA design for SynMultiplet Dataset and experimentally validated structures from PDB using RNAinformer.

| Test Set | Solved | Valid Seq. | Diversity | F1 | Seq. Rec. | NC |
|----------|--------|-----------|-----------|-----|-----------|-----|
| TS1 | 0.119±0.015 | 0.135±0.026 | 0.590±0.021 | 0.832±0.004 | 0.461±0.003 | 0.546±0.038 |
| TS2 | 0.214±0.015 | 0.150±0.035 | 0.575±0.011 | 0.923±0.004 | 0.479±0.007 | 0.751±0.006 |
| TS3 | 0.035±0.030 | 0.013±0.015 | 0.191±0.330 | 0.866±0.010 | 0.454±0.001 | 0.333±0.577 |
| TS-Hard | 0.167±0.041 | 0.081±0.032 | 0.558±0.037 | 0.834±0.007 | 0.465±0.006 | 0.433±0.111 |
| SynMultiplet | 0.089±0.006 | 0.035±0.008 | 0.712±0.001 | 0.862±0.002 | 0.432±0.001 | 0.421±0.038 |

Table 18: Results for RNA design for SynMultiplet Dataset and experimentally validated structures from PDB with target GC-content using RNAinformer.

| Test Set | Solved | Valid Seq. | Diversity | F1 | Seq. Rec | NC | GC-Error |
|----------|--------|-----------|-----------|-----|----------|-----|----------|
| TS1 | 0.114±0.017 | 0.093±0.005 | 0.656±0.032 | 0.834±0.004 | 0.470±0.004 | 0.648±0.070 | 0.005±0.001 |
| TS2 | 0.188±0.039 | 0.131±0.037 | 0.569±0.035 | 0.922±0.002 | 0.489±0.002 | 0.797±0.056 | 0.007±0.001 |
| TS3 | 0.018±0.030 | 0.007±0.012 | 0.176±0.305 | 0.858±0.002 | 0.462±0.007 | 0.167±0.289 | 0.004±0.001 |
| TS-Hard | 0.179±0.036 | 0.059±0.008 | 0.576±0.063 | 0.842±0.007 | 0.478±0.017 | 0.492±0.138 | 0.004±0.001 |
| SynMultiplet | 0.037±0.009 | 0.018±0.002 | 0.711±0.002 | 0.839±0.008 | 0.433±0.001 | 0.409±0.066 | 0.010±0.001 |

Table 19: Comparison between different versions of the RNAinformer pre-trained on synthetic or real-world data with and without finetuning on the experimentally validated structures from PDB. RNAinformerSyn refers to the RNAinformer model trained on synthetic data; RNAinformerSyn + FT refers to the model that was trained on synthetic data and finetuned with experimentally validated structures from PDB; RNAinformerNAT + FT refers to the model trained on existing (known) RNA secondary structures from publicly available sources and finetuned on experimentally validated structures from PDB.

| Test Set | RNAinformerSyn | | RNAinformerSyn + FT | | RNAinformerNAT + FT | |
|---|---|---|---|---|---|---|
| | F1 | Seq. Rec | F1 | Seq. Rec | F1 | Seq. Rec |
| TS1 | 0.832±0.004 | 0.461±0.003 | **0.861±0.003** | **0.520±0.005** | 0.346±0.022 | 0.427±0.005 |
| TS2 | 0.923±0.004 | 0.479±0.007 | **0.924±0.003** | **0.537±0.005** | 0.459±0.021 | 0.444±0.008 |
| TS3 | 0.866±0.010 | 0.454±0.001 | **0.877±0.005** | **0.500±0.015** | 0.324±0.004 | 0.413±0.014 |
| TS-Hard | 0.834±0.007 | 0.465±0.006 | **0.842±0.007** | **0.503±0.010** | 0.364±0.036 | 0.412±0.004 |

Table 20: Comparison of RNAinformer with and without finetuning on experimentally validated structures from PDB evaluated on the CASP15 RNA data.

| Test Set | RNAinformer | | RNAinformer + FT | |
|---|---|---|---|---|
| | F1 | MCC | F1 | MCC |
| CASP15 | **0.901±0.005** | **0.902±0.005** | 0.877±0.021 | 0.878±0.021 |

Table 21: Comparison between F1 Scores of designed sequences and PDB test set sequences using different folding algorithms.

| Folding Algo. | Sequence | TS1 | TS2 | TS3 | TS-Hard |
|---|---|---|---|---|---|
| RNAformer | Designed-Syn | **0.832** | **0.923** | **0.866** | **0.834** |
| | Designed-Nat | 0.346 | 0.459 | 0.324 | 0.364 |
| | PDB | 0.716 | 0.797 | 0.709 | 0.641 |
| SPOT-RNA | Designed-Syn | **0.719** | **0.824** | **0.731** | **0.677** |
| | Designed-Nat | 0.304 | 0.436 | 0.260 | 0.298 |
| | PDB | 0.714 | 0.800 | 0.671 | 0.663 |
| MXFold2 | Designed-Syn | **0.689** | **0.792** | **0.739** | 0.663 |
| | Designed-Nat | 0.269 | 0.406 | 0.197 | 0.242 |
| | PDB | 0.663 | 0.763 | 0.640 | **0.667** |
| UFold | Designed-Syn | 0.662 | 0.790 | **0.682** | **0.628** |
| | Designed-Nat | 0.256 | 0.410 | 0.219 | 0.246 |
| | PDB | **0.673** | **0.892** | 0.648 | 0.587 |

Table 22: Comparison of TM-scores of designed sequences and PDB test set sequences using AlphaFold3 for 3D structure predictions.

| Folding Algo. | Sequence | TS1 | TS2 | TS3 | TS-Hard |
|---|---|---|---|---|---|
| AlphaFold3 | Designed(Avg) | 0.331 | 0.289 | 0.325 | 0.280 |
| | Designed(Best) | 0.376 | 0.320 | 0.358 | 0.308 |
| | PDB | **0.537** | **0.354** | **0.500** | **0.410** |

Table 23: Difference in TM-scores of designed sequences and PDB test set sequences using AlphaFold3 for 3D structure predictions. We observe that RNAinformer predictions improve the TM score for orphan RNAs (where there is no MSA available for the ground truth sequence) but become worse for the sequences where MSA is available for the ground truth. Note that for all the designed sequences, AlphaFold did not find any MSA.

| Folding Algo. | MSA | TS1 | TS2 | TS3 | TS-Hard |
|---|---|---|---|---|---|
| AlphaFold3 | Orphan | 0.008 | -0.035 | 0.031 | 0.006 |
| | Non-Orphan | -0.224 | -0.033 | -0.188 | -0.197 |

Table 24: Evaluations of the designed sequences for the CASP15 data using AlphaFold 3 as the folding algorithm. We compare a version with and one without finetuning on known sequences (FT).

| Task | Id | FT | TM-score ↑ | RMSD ↓ |
|---|---|---|---|---|
| CPEB3_ribozyme (7QR4_1_B) | R1107 | no | **0.439** | **3.103** |
| | | yes | 0.391 | 3.507 |
| CPEB3 Ribozyme (7QR3_1_C) | R1108 | no | **0.373** | **3.490** |
| | | yes | 0.275 | 3.533 |
| CPEB3 Ribozyme (7QR3_1_D) | R1108 | no | **0.311** | **3.250** |
| | | yes | 0.307 | 3.327 |
| Cloverleaf RNA (8S95_1_C) | R1116 | no | 0.357 | 4.797 |
| | | yes | **0.395** | **4.050** |
| SARS-CoV-2 SL5 (8UYS_1_A) | R1149 | no | 0.325 | **3.640** |
| | | yes | **0.329** | 4.003 |
| BtCoV-HKU5 SL5 (8UYE_1_A) | R1156 | no | **0.379** | **3.803** |
| | | yes | 0.312 | 3.867 |
| BtCoV-HKU5 SL5 (8UYG_1_A) | R1156 | no | 0.326 | **3.743** |
| | | yes | **0.352** | 4.293 |
| BtCoV-HKU5 SL5 (8UYJ_1_A) | R1156 | no | 0.313 | **3.833** |
| | | yes | **0.314** | 4.090 |
| A-6B (7YR7_1_A) | R1189 | no | **0.259** | **4.003** |
| | | yes | 0.240 | 4.167 |
| A-4B (7YR6_1_A) | R1190 | no | 0.251 | 3.503 |
| | | yes | **0.270** | **2.873** |

