# OpenReview forum: "RNAinformer: Generative RNA Design with Tertiary Interactions"
_ICLR.cc/2025/Conference — Submitted to ICLR 2025_

### Official Review · Reviewer_VnQq · 2024-11-02

**Soundness:** 2
**Presentation:** 3
**Contribution:** 2
**Rating:** 5
**Confidence:** 4

**Summary:**

This manuscript explores a more general RNA inverse folding problem, beyond the canonical base-pairing constraint. The authors propose a Transformer-based learning model, leveraging the adjacency matrix to capture more intricate nucleotide tertiary interactions and map them to sequences. Comprehensive experiments are conducted to validate the effectiveness of this model.

**Strengths:**

1.	This paper addresses the limitations of traditional dot-bracket secondary structure inverse folding by exploring adjacent-matrix based methods to represent more complex structural motifs. This approach aligns more closely with the practical requirements of functional RNA design.
2.	In the absence of prior research on this topic, this paper constructs extensive training data and diverse test tasks, demonstrating the capacities of the proposed model in multiple aspects.
3.	The overall writing structure of this paper is clear and logically organized, making it easy to follow.

**Weaknesses:**

1.	The related work in this paper is insufficient. It overlooks a class of RNA inverse folding methods based on tertiary structure, such as RDesign [1] and RiboDiffusion [2], which directly use the RNA tertiary structure backbone as model input and implicitly model structural interactions like pseudoknots. These methods are highly relevant to the topic of the paper.

2.	The benchmarking in this paper has several flaws:
    (1)	The folding-back algorithm is overly simplistic and lacks diversity. Tertiary structure prediction methods should be included to assess whether the designed sequences meet expected tertiary interactions. Additionally, the errors caused by the folding-back algorithm are not adequately explained.
    (2)	The model's performance under novel structures, i.e., samples with significant differences from the training set, is not analyzed.
(3)	There are relatively few baseline methods.  For adjacency matrices, graph neural network-based methods and ResNet-based methods are commonly used to represent RNA data and could be easily adapted to the current task.
(4)	Although it is reasonable to use synthetic data for proof of concept, using it as the core test set is far from the actual application scenario and may introduce more unexpected errors.
(5)	There is a lack of comparison on public benchmarks, such as Eterna100, which is crucial to demonstrate the basic capabilities of the model.

3.	The paper lacks case study results to demonstrate the correctness, rationality, and novelty of the designed sequences.

4.	For the machine learning community, the method in this paper adopts a well-studied Transformer structure and classical loss function, making it difficult to claim a technical contribution. For the biology community, the paper lacks rigorous and comprehensive evaluation to demonstrate its advantages and reliability. Overall, this manuscript does not yet meet the standard for publication.

[1] RDesign: Hierarchical Data-efficient Representation Learning for Tertiary Structure-based RNA Design. The Twelfth International Conference on Learning Representations. 2024.

[2] RiboDiffusion: tertiary structure-based RNA inverse folding with generative diffusion models. Bioinformatics 40. Supplement_1 (2024): i347-i356.

**Questions:**

Please refer to the Weakness section for more details.
1. Can some GNN-based baseline methods be added?
2. Can benchmarking be done on some public test sets, such as Eterna100 and Openknot?
3. Can the model design performance on relatively novel structures be tested on natural RNA of CASP15?
4. Can the designed sequence be folded using the tertiary structure prediction model?

---

> ### Author Response · Authors · 2024-11-25
> **Author response to Reviewer VnQq**
>
> Dear Reviewer VnQq,
>
> We thank you for your valuable feedback and for pointing out the novel aspects of our work and the comprehensive analysis. We will address your concerns and questions in detail in the following.
>
> >The related work in this paper is insufficient. It overlooks a class of RNA inverse folding methods based on tertiary structure, such as RDesign [1] and RiboDiffusion [2], which directly use the RNA tertiary structure backbone as model input and implicitly model structural interactions like pseudoknots. These methods are highly relevant to the topic of the paper.
>
> We thank the reviewer for this useful comment. We have added a section on 3D RNA design algorithms to the Related Work section in the revised manuscript.
>
> >The benchmarking in this paper has several flaws: (1) The folding-back algorithm is overly simplistic and lacks diversity. Tertiary structure prediction methods should be included to assess whether the designed sequences meet expected tertiary interactions. Additionally, the errors caused by the folding-back algorithm are not adequately explained. AND Can the designed sequence be folded using the tertiary structure prediction model?
>
> We thank the reviewer for this suggestion. We folded the predictions for the PDB test sets using AlphaFold 3 (AF3). The results are shown in Tables 22 and 23. We find that RNAinformer can improve the design for orphan RNAs, where there is no MSA available for the ground truth data.
>
> >The model's performance under novel structures, i.e., samples with significant differences from the training set, is not analyzed. AND Can the model design performance on relatively novel structures be tested on natural RNA of CASP15?
>
> We again would like to thank the reviewer for the useful suggestion. We add an analysis of predictions on the CASP15 RNA only data to the revised version of our manuscript. The results are shown in Table 24.  We use these results to assess the performance of RNAInformer with and without finetuning on known RNAs from PDB. Surprisingly, we observe that the RNAinformer trained on synthetic data only achieves better results compared to the version that was finetuned on real-world data.
>
> >There are relatively few baseline methods. For adjacency matrices, graph neural network-based methods and ResNet-based methods are commonly used to represent RNA data and could be easily adapted to the current task. AND Can some GNN-based baseline methods be added?
>
> We add a GNN baseline based on the structTransformer provided by [1]. We run the baseline with the same batch size and number of steps as the RNAinformer. The results are shown in Table 16.
>
> >Although it is reasonable to use synthetic data for proof of concept, using it as the core test set is far from the actual application scenario and may introduce more unexpected errors.
>
> We agree with the reviewer that evaluation on synthetic data alone is not the way to go when training on synthetic data. However, we disagree with the reviewer that we evaluate on the synthetic data only. We run experiments on four test sets of experimentally validated structures from the PDB. These sets were initially provided by [2] and [3] and are commonly used for evaluations of deep learning methods in the field of RNA secondary structure prediction.
>
> Furthermore, to assess the level of overfitting to the folding engine that was used during the generation of synthetic data, we fold all generated sequences with three more folding algorithms from the literature. The results are shown in Table 21 of the revised version of our manuscript (Table 19 in our initial submission). We find that RNAinformer does not overfit the folding engine used during synthetic data generation but that it generates sequences that improve the F1 score compared to folding the ground truth sequence across most of the datasets and with different folding engines. We call this “Improved foldability” in the manuscript. We further extend this analysis to include AlphaFold 3 as a 3D folding algorithm and observe similar results for orphan RNAs, but not for RNAs where MSA is available for the ground truth data. The respective results are shown in Tables 22 and 23 of the revised manuscript.
>
> Finally, we would like to emphasize that we also evaluate the RNAinformer on a realistic task of riboswitch design in Section 5.3.

---

> ### Author Response · Authors · 2024-11-25
> **Author response to Reviewer VnQq continued**
>
> >There is a lack of comparison on public benchmarks, such as Eterna100, which is crucial to demonstrate the basic capabilities of the model. AND Can benchmarking be done on some public test sets, such as Eterna100 and Openknot?
>
> We agree with the reviewer that public benchmarks could provide valuable insights about strengths and weaknesses of a new algorithm. However, as also stated by the reviewer in the initial Review, the main focus of our work is the design of RNAs for all sorts of base pairs. This includes non-canonical base pairs, pseudoknots and base multiplets. The Eterna100 benchmark and the OpenKnot benchmark both do only provide nested structures or pseudoknotted structures, but not base multiplets. We, therefore, use existing and commonly used public benchmarks from the RNA secondary structure prediction literature instead of creating a new benchmark dataset for RNA design. For our evaluations on nested structures, we already use an additional, more practically relevant benchmark of theophylline riboswitch design. Furthermore, evaluations on the Eterna100 benchmark typically involve 24 hour runs on each of the benchmark tasks, refining the predictions with every iteration. In contrast, we only generate 20 samples with RNAinformer for all experiments on nested structures.
>
> We again thank the reviewer for the useful feedback. We hope that we addressed all questions and concerns but we are happy to answer any further questions if necessary. If there are no further concerns, we would like to kindly ask the reviewer to reconsider our score.
>
> With kind regards,
>
> The Authors

---

> ### Comment · Reviewer_VnQq · 2024-11-27
>
> Thanks for the author's response. I will increase the score to 5. The main reasons preventing the score from improving further are the relatively limited innovations in the model, the lack of adequate baseline comparisons on new tasks, and the absence of convincing generation case studies.

---

### Official Review · Reviewer_WYDo · 2024-11-02

**Soundness:** 4
**Presentation:** 4
**Contribution:** 3
**Rating:** 6
**Confidence:** 4

**Summary:**

The authors present RNAInformer, an RNA secondary structure inverse folding method. The authors identify that there has been strong recent progress in RNA secondary structure prediction, and use this as the motivation for the work. The authors additionally identify that the vocabulary of base-base interactions in RNA secondary structure is much larger than what the community typically models and account for this in their method through an axial attention mechanism

**Strengths:**

Overall I think the work is quite carefully considered: from the problem statement to the dataset setup and splits, and the evaluation. The empirical results are strong, across the different design tasks and the authors demonstrate the ability to incorporate additional property constraints (GC content) convincingly. The paper is laid out clearly and well-written.

**Weaknesses:**

The predominant weakness in this work is the modest technical novelty in the proposed architecture. This reviewer is of the opinion that sound and well-executed applied work (such as this) has a place in venues such as ICLR but the judgment on this rests with the AC.

**Questions:**

* While the hyperparameters are described, there is little discussion of how they were obtained; how much sweeping did the authors perform?

* Do the authors give any consideration to non-canonical/modified nucleotides?

* Could the authors share some reasoning why the model trained on synthetic data performs better than the finetuned model?

* The pairwise hamming distance for diversity is designed to account for base pair flips?

---

> ### Author Response · Authors · 2024-11-25
> **Author response to Reviewer WYDo**
>
> Dear Reviewer WYDo,
>
> We thank you for the valuable feedback. Specifically, we acknowledge the reviewers’ comments that our work is well-executed. We will address the individual questions in the following.
>
> >While the hyperparameters are described, there is little discussion of how they were obtained; how much sweeping did the authors perform?
>
> Due to the number of experimental settings, we manually tuned the model using our validation sets to obtain a good common model setting across our evaluations. We mainly tuned the learning rate for our batch size as we found it to be the most important hyperparameter. The training stability and validation performance depended on a fast initial lowering of the training loss.
>
> >Do the authors give any consideration to non-canonical/modified nucleotides?
>
> We thank the reviewer for this valuable comment but have to admit that we do not include modified nucleotides.
> However, we are also not aware of any secondary structure prediction algorithm that is capable of handling modified nucleotides.
>
> >Could the authors share some reasoning why the model trained on synthetic data performs better than the finetuned model?
>
> We think that the reason is the limited availability of data containing all kinds of base pairs.
>
> While there exists a lot of data from comparative sequence analysis and predicted secondary structures in the public domain, this data typically lacks base multiplets and often also pseudoknots. The only reliable source of data with all kinds of base pairs is the PDB, where RNA only data describes a very small fraction of all 3D structures.
> During training on known RNAs, we thus rarely visit sequences with multiplets (or always the same when oversampling multiplet data). With synthetic data, on the other hand, we can generate a lot more samples that contain all kinds of base pairs. Given that our folding algorithm is strong enough (we use RNAformer which appears to be the state-of-the-art folding algorithm), it seems that we can learn these kinds of features much better.
>
> >The pairwise hamming distance for diversity is designed to account for base pair flips?
>
> No, the pairwise hamming distance metric doesn't take into consideration base pair flips. A generated sequence that contains a base pair flip is considered a different sequence. However, we do not observe a tendency of RNAinformer to only introduce base pair flips.
>
> We thank the reviewer again for the valuable feedback. If there are any further questions we are happy to answer them. Otherwise, we would appreciate it if the reviewer would consider updating our score.
>
> With kind regards,
>
> The Authors

---

> > ### Comment · Reviewer_WYDo · 2024-11-27
> > **Rebuttal acknowledged**
> >
> > I thank the authors for addressing my questions.
> >
> > I think my original score is appropriate and I will therefore maintain it. Justification: the next scoring option available to me as a reviewer would be an 8 and that would require greater technical novelty for me to deem it appropriate.

---

### Official Review · Reviewer_Lp2j · 2024-11-03

**Soundness:** 3
**Presentation:** 2
**Contribution:** 2
**Rating:** 5
**Confidence:** 4

**Summary:**

The authors propose RNAinformer, a new method for solving the problem of inverse RNA design. This problem considers finding the nucleotide sequence of RNAs that will result in the desired structure once they fold. Authors approach this problem in a more general setting by considering pseudoknots and base multiplets. They train a conditional transformer model to generate RNA sequences conditioned on the desired GC content and possibly additional constraints. They also introduce a new pipeline for generating synthetic data that can be used for generative models.

**Strengths:**

Authors have invested time to design a pipeline for generating synthetic data that can become useful for designing learning-based methods since the experimentally verified sequences are more costly to generate.

Authors consider the problem in the general form where different types of base pairing (e.g., non-canonical base-pairs and pseudoknots) exist, and also allow additional constraints (e.g., GC content) to be specified. The problem is highly important and good solutions can be very impactful.

**Weaknesses:**

There are many recent methods for solving the problem of inverse RNA design which have not been used for comparisons in this paper. The comparisons are limited to only a few of the prior works (mostly only 2). Some of these prior methods have similarly used conditional autoregressive formulation similar to this work. The paper in [1] targets an even more general problem by inverse RNA tertiary structure which solves the RNA inverse problem as a subproblem. Without thorough comparisons it is hard to ensure the merits of the newly proposed approach.


1. Tan, C., Zhang, Y., Gao, Z., Hu, B., Li, S., Liu, Z., & Li, S. Z. (2024). RDesign: Hierarchical Data-efficient Representation Learning for Tertiary Structure-based RNA Design. In The Twelfth International Conference on Learning Representations.

2. Rubio-Largo, Á., Lozano-García, N., Granado-Criado, J. M., & Vega-Rodríguez, M. A. (2023). Solving the RNA inverse folding problem through target structure decomposition and Multiobjective Evolutionary Computation. Applied Soft Computing, 110779.

3. Merleau, N. S., & Smerlak, M. (2022). aRNAque: an evolutionary algorithm for inverse pseudoknotted RNA folding inspired by Lévy flights. BMC bioinformatics, 23(1), 335.


There is also a similarity between the protein inverse folding problem and RNA inverse folding and since the former has been studied more, most models can still be applicable to the RNAs (instead of 20 enzymes we will have 4 nucleotides). Therefore, works in RNA inverse folding still use these methods as baselines. For example the following works are used as baselines for RNA inverse problems as well.

4. Ingraham, J., Garg, V., Barzilay, R., & Jaakkola, T. (2019). Generative models for graph-based protein design. Advances in neural information processing systems, 32.

5. Gao, Z., Tan, C., & Li, S. Z.(2023) PiFold: Toward effective and efficient protein inverse folding. In The Eleventh International Conference on Learning Representations.


For many of the comparisons, such as the ones on nested structures in which the proposed method achieves significantly worse results than the baseline methods (SAMFEO), the authors justify the presented results by highlighting the diversity of their generated outcomes. However, the first goal is to derive valid sequences. If neither of the sequences for a task is not valid what would be the benefit of diversity?

That being said, even for most other prior methods, we can still tune the diversity. For example you can change the temperature value in the Boltzmann sampling part of SAMFEO to get a less skewed distribution that leads to a higher diversity. However, it is important to have a skewed enough distribution to make sure we have a high recall among a small number of candidates.

The method authors use for measuring the diversity of the valid sequences is computing the mean of pairwise hamming distances between the sequences. This may lead to an overestimation of the diversity because if two RNA sequences have just an indel (unpaired nucleotide) compared to each other, it considers all the subsequence bases as different. I think in this case something similar to the Needleman-Wunsch algorithm would be a better measure of dissimilarity.

It is also not very clear why diversity is an important factor here because unlike image generation and text generation, for RNA design it is more important to get the correct design among the few samples. For example, when looking for a specific RNA to bind to the snoRNA of a new virus to inhibit its activity, why should one care about the diversity of the proposed solutions rather than just having the top few solutions that are more likely to work. It is more important to have the expected RNA among the top few generated solutions to also decrease the costs of downstream in-vivo experiments.

Some results such as the one presented in Figure 4 might portray the amount of non-canonical base-pairs (NC) as a measure that has to be maximized. However, although these non-canonical base-pairs are present in folded RNAs, this amount has to be the correct amount. If these values are maximized by the model but does not lead to valid sequences then why should it be considered as a good thing?

For experimentally verified 3D structures, which can be considered the gold standard set for evaluation, authors only report the results of their method and do not compare it to any other baseline. From table 17, the ratio of the valid sequences seems to be very low and it would be important to see if prior methods do not outperform RNAinformer on this dataset.

For the experiments on RNA design with pseudoknots authors only compare their method to antaRNA which is a relatively old method (2015) while many newer methods with superior results have been published since then.

The length of the supported sequences is mentioned to be capped at 200 in the experiments. That is relatively low to what has been done in recently proposed methods (e.g., 900 in SAMFEO and 500 nucleotides in RDESIGN -- see [1] above). This is an important shortcoming because it is well-known that with the increased length of RNA the problem becomes more difficult. Authors have mentioned this in their Limitations section, but they mention that it is enough for current benchmarks while there are other benchmarks that contain RNAs with more nucleotides and they are simply not used in the presented experiments.

While authors propose a new pipeline for designing their synthetic datasets, it would have been better to use the datasets used by prior methods (e.g., Eterna100 or the one curated in [1]) to show the performance of the presented method in comparison to the baseline methods, and then mention the motivation behind using a new pipeline for generating a dataset.

**Questions:**

Please see the weaknesses.

---

> ### Author Response · Authors · 2024-11-25
> **Author response to Reviewer Lp2j**
>
> Dear Reviewer Lp2j,
>
> We thank you for the valuable feedback and for highlighting the usefulness of our data generating pipelines for future work and the importance of the problem in general.
> We will address your questions and concerns in detail in the following.
>
> >There are many recent methods for solving the problem of inverse RNA design which have not been used for comparisons in this paper. The comparisons are limited to only a few of the prior works (mostly only 2). Some of these prior methods have similarly used conditional autoregressive formulation similar to this work. The paper in [1] targets an even more general problem by inverse RNA tertiary structure which solves the RNA inverse problem as a subproblem. Without thorough comparisons it is hard to ensure the merits of the newly proposed approach.
>
> We thank the reviewer for sharing these references with us. We agree that there are methods available that we did not consider for evaluation. Regarding the three works suggested by the reviewer, we did not include these methods for different reasons.
>
> Regarding [1]: While RDesign appears to be a promising approach, the algorithm tackles a different problem than RNAinformer, RNA design based on 3D structure information. This approach poses different challenges on the algorithm compared to pure secondary structure based design. Furthermore, the algorithm requires additional 3D structure information for the design of RNAs which is often not available in real-world scenarios. Finally, secondary structures designed by RDesign are limited to canonical base interactions in dot-bracket notation. We, therefore, exclude RDesign from our evaluations.
>
> Regarding [2]: The described algorithm designs RNAs for nested structures in dot-bracket notation only. Since we are mainly interested in designing RNAs for all possible base interactions based on matrix representations, we exclude it from our evaluations.
>
> Regarding [3]: We tried to run aRNAque on our datasets. However, due to long runtimes for each evaluation by arNAque and because aRNAque doesn't work with level 3 pseudoknots, which antarna can manage and which are also in our test set, we did not include aRNAque in our evaluations.
>
> >There is also a similarity between the protein inverse folding problem and RNA inverse folding and since the former has been studied more, most models can still be applicable to the RNAs (instead of 20 enzymes we will have 4 nucleotides). Therefore, works in RNA inverse folding still use these methods as baselines. For example the following works are used as baselines for RNA inverse problems as well.
>
> We add a GNN baseline based on the structTransformer provided by [1]. We run the baseline with the same batch size and number of steps as the RNAinformer. The results are shown in Table 16 of the revised version of our manuscript.
>
> >For many of the comparisons, such as the ones on nested structures in which the proposed method achieves significantly worse results than the baseline methods (SAMFEO), the authors justify the presented results by highlighting the diversity of their generated outcomes. However, the first goal is to derive valid sequences. If neither of the sequences for a task is not valid what would be the benefit of diversity?
>
> We agree with the reviewer that we do not achieve SOTA performance for this explicit experiment (while still achieving second best performance, solving 91% of the tasks while the third best competitor achieves 77% only). We think that we also clearly state this in the text as mentioned by the reviewer. However, we disagree that this is the case for many comparisons, in fact, this is the only experiment where RNAinformer is outperformed by any of the reported (specialized) baselines.
>
> Regarding diversity, we would like to clarify that it is calculated for all the designs that solve a given structure (valid sequences). We clearly observe that the generated solutions of RNAinformer are more diverse than those of SAMFEO and think that this is a very important result for an auto-regressive modeling approach trained on sequence recovery while only generating 20 samples per task.
>
> Overall, we think that reporting cases where our method fails (or does not achieve SOTA results) is important and could lead to a better understanding of the strengths and weaknesses and thereby to better methods in the future.

---

> ### Author Response · Authors · 2024-11-25
> **Author response to Reviewer Lp2j continued**
>
> > That being said, even for most other prior methods, we can still tune the diversity. For example you can change the temperature value in the Boltzmann sampling part of SAMFEO to get a less skewed distribution that leads to a higher diversity. However, it is important to have a skewed enough distribution to make sure we have a high recall among a small number of candidates.
>
> We think the RNAinformer shows strong performance in terms of solved tasks and high diversity. While diversity might be increased artificially, at least for RNAinformer there is no need to do so. In contrast, we think that our model shows strong performance across all tasks, while ensuring high diversity of the generated candidates.
>
> >The method authors use for measuring the diversity of the valid sequences is computing the mean of pairwise hamming distances between the sequences. This may lead to an overestimation of the diversity because if two RNA sequences have just an indel (unpaired nucleotide) compared to each other, it considers all the subsequence bases as different. I think in this case something similar to the Needleman-Wunsch algorithm would be a better measure of dissimilarity.
>
> For our calculations of diversity, we only consider sequences of the same length that were generated for a given target (generation is length limited). Therefore, aligning sequences e.g. using Needleman-Wunsch is not necessary (there are no indels).
>
> >It is also not very clear why diversity is an important factor here because unlike image generation and text generation, for RNA design it is more important to get the correct design among the few samples. For example, when looking for a specific RNA to bind to the snoRNA of a new virus to inhibit its activity, why should one care about the diversity of the proposed solutions rather than just having the top few solutions that are more likely to work. It is more important to have the expected RNA among the top few generated solutions to also decrease the costs of downstream in-vivo experiments.
>
> We think that In silico RNA design methods should support experimentalists during the search for promising candidates that can then be subsequently analyzed in the wet-lab. It is, therefore, important to provide a large list of diverse and promising candidates for screening.
> This is in line with recent developments and publications in the RNA design community [6,7]. To be maximally useful, an RNA design algorithm should be able to generate as many valid sequences which are also as diverse as possible to explore the potential space of solutions and reduce human effort.
>
> [6] Hammer, S., Günzel, C., Mörl, M., & Findeiß, S. (2019). Evolving methods for rational de novo design of functional RNA molecules. Methods, 161, 54-63.
>
> [7] Runge, F., Franke, J., Fertmann, D., Backofen, R., & Hutter, F. (2024). Partial RNA design. Bioinformatics, 40(Supplement_1), i437-i445.
>
> >Some results such as the one presented in Figure 4 might portray the amount of non-canonical base-pairs (NC) as a measure that has to be maximized. However, although these non-canonical base-pairs are present in folded RNAs, this amount has to be the correct amount. If these values are maximized by the model but does not lead to valid sequences then why should it be considered as a good thing?
>
> We think this part requires clarification:
>
> We are not aiming at maximizing the number of non-canonical base pairs but report the number of valid sequences that contain non-canonical base pairs. The reason is that existing secondary structure based RNA design methods cannot design RNAs that contain non-canonical base pairs at all – we show that the designs of RNAinformer contain non-canonical base pairs in many cases, similar to the ground truth data.
>
> >For experimentally verified 3D structures, which can be considered the gold standard set for evaluation, authors only report the results of their method and do not compare it to any other baseline. From table 17, the ratio of the valid sequences seems to be very low and it would be important to see if prior methods do not outperform RNAinformer on this dataset.
>
> We agree with the reviewer that we do not compare to any other method on the experimentally validated structures. The reason is that RNAinformer is the first secondary structure based RNA design algorithm that is capable of designing RNAs for secondary structures that contain all kinds of base pairs. However, we implement a GNN baseline and add the results to Table 16 in the Appendix.
>
> We also agree with the reviewer that the ratio of valid sequences is relatively low. However, this is a pioneering work that – for the first time – enables the design of RNAs with all kinds of base interactions and we show that we can solve some of the tasks that contain pseudoknots, multiplets and non-canonical base pairs. This is novel in the field of RNA design.

---

> ### Author Response · Authors · 2024-11-25
> **Author response to Reviewer Lp2j continued**
>
> >For the experiments on RNA design with pseudoknots authors only compare their method to antaRNA which is a relatively old method (2015) while many newer methods with superior results have been published since then.
>
> We agree with the reviewer that antaRNA is not new. However, it is still one of the main methods used for pseudoknotted RNA design. That being said, we also tried to evaluate aRNAque on the dataset but the runtimes were unexpectedly high and because aRNAque doesn't work with level 3 pseudoknots which antarna can handle, we skipped further evaluations of aRNAque.
>
> >The length of the supported sequences is mentioned to be capped at 200 in the experiments. That is relatively low to what has been done in recently proposed methods (e.g., 900 in SAMFEO and 500 nucleotides in RDESIGN -- see [1] above). This is an important shortcoming because it is well-known that with the increased length of RNA the problem becomes more difficult. Authors have mentioned this in their Limitations section, but they mention that it is enough for current benchmarks while there are other benchmarks that contain RNAs with more nucleotides and they are simply not used in the presented experiments.
>
> We agree with the reviewer that this is a major limitation of RNAinformer that we also clearly state in the limitations section in the concluding remarks. We also agree that structure prediction for  longer sequences is typically more challenging and in turn also poses challenges to design algorithms.
> However, most of the experimentally validated sequences depicted in the PDB are relatively short. For instance, the most commonly used PDB data sets used for the evaluation of deep learning approaches (TS1, TS2, TS3, and TS-Hard; which we also use here) have a maximum sequence length of 189nt. Similarly, from the originally collected data from PDB (collected from RNAsolo which essentially is a PDB mirror) in the RDesign publication ([1]), 87% are shorter than 100nt. The limitation to 200nt thus is unfortunate, but we still think that it captures most of the experimentally validated samples in the available benchmarks.
>
> >While authors propose a new pipeline for designing their synthetic datasets, it would have been better to use the datasets used by prior methods (e.g., Eterna100 or the one curated in [1]) to show the performance of the presented method in comparison to the baseline methods, and then mention the motivation behind using a new pipeline for generating a dataset.
>
> We do not fully agree with the reviewer here. While we agree that it is always interesting to see comparisons to existing methods, RNAinformer is the only available method for secondary structure based design that can tackle real-world structures from PDB. Furthermore, we think that using synthetic data is highly desirable, particularly in the biological domain, and that our results show that we can transfer well from synthetic to real world examples. We think that this result could also be of interest to the community. The correct preparation of the synthetic data is one of the key steps to reliably assess the performance on the real data and to avoid data leakage issues. We, therefore, carefully prepared our synthetic pipelines to make them a valuable starting point for future research.
>
> We thank the reviewer again for the valuable feedback. We hope that we adequately addressed all questions and concerns. However, if there are still any questions left, we are happy to answer them! If we have answered your questions satisfactorily, we would appreciate it if you would consider increasing our score.
>
> With kind regards,
>
> The Authors

---

> > ### Comment · Reviewer_Lp2j · 2024-11-28
> >
> > I would like to thank the authors for their responses! The problem that the authors have targeted in their work is indeed an important problem; however, I think since the introduced model does not have a novel component it should at least present a comprehensive comparison to other base-lines proving its effectiveness. Although the introduced model has additional capabilities (e.g., including pseudo-knots), I still expect to see it outperform other methods in well-known benchmark datasets that have been used by prior work, even though they are designed for simpler evaluations because they can be considered subproblems (e.g., finding the sequences without desired tertiary structure that doesn't contain a pseudoknot).
> >
> > I have two other minor follow-up comments regarding the provided responses:
> >
> > - Regarding the capability of other models such as SAMFEO for generating more diverse sequences, they mentioned that "While diversity might be increased artificially, at least for RNAinformer there is no need to do so." However, changing the temperature for the sampling from any generative model is an inherent hyperparameter. There is nothing artificial about it. The same hyper-parameters and strategies are used for other generative models such as transformers used by the authors and can be tuned for a desired level of diversity.
> >
> > - Regarding the added results for the GNN baseline, the results seem to be the same as random. Do the authors explore different hyperparameters and settings to generate the best possible results?
> >
> > Still, the major concern I raised at the beginning of this comment prevents me from raising my score; however, I would remain open to discussion with other reviewers and AC and would not challenge it if they decide to accept the paper.

---

### Official Review · Reviewer_DDzf · 2024-11-04

**Soundness:** 2
**Presentation:** 3
**Contribution:** 3
**Rating:** 5
**Confidence:** 4

**Summary:**

### A method for 2d structure based RNA inverse folding enabling - for the first time - arbitrary interaction types (e.g. pseudo-knots, non-canonical base pairs, ...)

This paper introduces a secondary structure based RNA inverse folding model (RNAinformer) that is capable of designing RNA sequences from secondary structures with arbitrary interaction types (e.g. non-canonical base pairs, pseudo knots, base multiplets) that were not representable in previous 2D based inverse folding methods. This improvement of being able to represent these arbitrary interaction types is achieved by working with the more expressive adjacency matrix representation instead of dot-bracket representations of the secondary structure.

The RNAinformer model is based on an auto-regressive encoder-decoder transformer. The secondary structure (in the form of an adjacency matrix) is encoded via axial attention (similar to the RNAformer structure prediction model) and finally pooled from a 2d to a 1d vector that is passed to the decoder for decoding into an RNA sequence. RNAinformer also supports constrained generation based on masked sequences, which are embedded into a 2d representation by the encoder if provided, or desired GC content (linearly embedded and added to the embedding).

The authors also make an interesting, strong claim that training on synthetic data only improves performance over training with experimental data, that -- if true -- would be of significant interest to the community.

**Strengths:**

1. The paper is well written, clear and has a nice flow.

2. The provided codebase looks well structured & documented upon a first spot check.

3. I'm a big fan of your spider plots showing a variety of metrics of interest (valid sequences, diversity, solved ,...)

4. The performance of the model and its enhanced design capabilities (GC content, masking, leveraging structure beyond what can be represented in simple dot-brackets) are promising and of interest to the community. If the authors can demonstrate that these also hold when using non-synthetic training data or further support their claim on the synthetic training data being superior to currently available experimental training data this would be of much interest to the community. (c.f. also weaknesses).

**Weaknesses:**

1. A link to the 3D inverse folding literature is currently missing: In the related work section I would have expected to see a mention of the (deep learning based) inverse folding efforts based on 3D structure (e.g. Rosetta, gRNADe, etc.). How does secondary structure-based inverse folding perform compared to 3D based inverse folding, in cases where 3D structure is available? This may represent a route of further testing and strengthening the hypothesis that the essential information for rna structure-function relations are encoded in 2D connectivity patterns (base pairs, psuedo-knots, base multiplets, ...).

2. I am somewhat concerned by the purely synthetic data based training strategy, since synthetic data are created with the same method that is used for evaluation. The authors argue that this allows side-stepping the data gap in gold standard secondary structure data, which is limited in the PDB. However, ultimately any secondary structure prediction method will have been trained on some level of structural data, partially exhibiting the biases and limitations of the datasets that the authors discussed. In addition to those, the models might also have certain model-specific biases that the inverse folding model may then pick up. Since the evaluation is also done by the same prediction model (e.g. RNAfold in 5.1) this risks reinforcing those model-specific biases. While the authors provide an experiment in the appendix to address this, I believe this point deserves further discussion and should be featured more central stage (at the very least including table 20, and possibly a test also for some of the other tasks specifically). This point is quite interesting and -- if true and well supported -- could be of significant interest to the community beyond the method in this paper alone.
Some analyses I would like to see in this regard:
- a comparison of training on synthetic data from model 1, but an evaluation with another, independent structure prediction model 2.
- an analysis of the sequence recovery (an imperfect metric, I know, but still somewhat informative to exclude cases of the inverse folding model overfitting on quirks of the structure prediction model). If you were to go all out, an evolutionary inspired recovery may be even more informative (c.f. https://openreview.net/forum?id=y5L8W0KRUX&referrer=%5Bthe%20profile%20of%20Chengyue%20Gong%5D(%2Fprofile%3Fid%3D~Chengyue_Gong1) in the protein world

**Questions:**

- In Table 1, it'd be helpful if the authors could highlight what % means (here % solved), as well as the n=... that was used for this estimate and the topk from which the answer was picked. From the description, it is not quite clear to me how those % were obtained.

- Were all benchmarked methods re-trained on the same synthetic data?  (for each of the tasks)

---

> ### Author Response · Authors · 2024-11-25
> **Author response to Reviewer DDzf**
>
> Dear Reviewer DDzf,
>
> We thank you for your valuable feedback and for acknowledging the style of writing and the quality of our codebase. We also appreciate the comment on our spider plots and for assessing our work as relevant for the community. In the following, we address your questions and concerns in detail.
>
> >A link to the 3D inverse folding literature is currently missing: In the related work section I would have expected to see a mention of the (deep learning based) inverse folding efforts based on 3D structure (e.g. Rosetta, gRNADe, etc.). How does secondary structure-based inverse folding perform compared to 3D based inverse folding, in cases where 3D structure is available? This may represent a route of further testing and strengthening the hypothesis that the essential information for rna structure-function relations are encoded in 2D connectivity patterns (base pairs, psuedo-knots, base multiplets, ...).
>
> We thank the reviewer for this helpful comment. We add a discussion about 3D RNA design methods in the Related Work section of the revised version of our manuscript.
> We also agree with the reviewer that comparing the performance of 3D methods with 2D design approaches could lead to interesting insights. We will try to obtain predictions for our test data using dedicated 3D RNA design approaches in the future.
>
> >I am somewhat concerned by the purely synthetic data based training strategy, since synthetic data are created with the same method that is used for evaluation. However, ultimately any secondary structure prediction method will have been trained on some level of structural data, partially exhibiting the biases and limitations of the datasets that the authors discussed. In addition to those, the models might also have certain model-specific biases that the inverse folding model may then pick up. Since the evaluation is also done by the same prediction model (e.g. RNAfold in 5.1) this risks reinforcing those model-specific biases.
>
> Generally, we agree with the reviewer that RNA design is tightly connected to RNA folding as we always require a folding oracle for the evaluation of a given design as long as there is no lab-in-the-loop approach involved. In this regard, different design methods use different folding oracles (most of them use RNAfold) and we would like to emphasize that one of our major contributions is that our approach allows us to employ any RNA secondary structure prediction algorithms. We also agree with the reviewer that this procedure bears the risk of overfitting the folding algorithm, as recently shown in a benchmarking paper for learning-based approaches for RNA design [1]. Therefore, we analyze the predictions of RNAinformer using multiple state-of-the-art deep learning based folding algorithms (SPOT-RNA, MXFold2, UFold). The results are shown in Table 19 of our initial submission (or Table 21 of our revised manuscript) and indicate that RNAinformer does not overfit the RNAformer model, but seems to design RNAs with improved foldability for nearly every folding engine compared to the original PDB sequences across the test sets.
>
> To further investigate this, we use AlphaFold 3 to predict 3D structures for the generated sequences and analyze these in terms of TM Score. Our results (Tables 22 and 23 in the revised manuscript) indicate that RNAinformer predictions improve the predictions in a fair comparison on orphan RNAs but due to a lack in MSA for the generated sequences, show strongly decreased performance on sequences where there is MSA available for the ground truth data.
>
> [1] Koodli, R. V., Rudolfs, B., Wayment-Steele, H. K., Eterna Structure Designers, & Das, R. (2021). Redesigning the EteRNA100 for the Vienna 2 folding engine. BioRxiv, 2021-08.
>
> >While the authors provide an experiment in the appendix to address this, I believe this point deserves further discussion and should be featured more central stage (at the very least including table 20, and possibly a test also for some of the other tasks specifically). This point is quite interesting and -- if true and well supported -- could be of significant interest to the community beyond the method in this paper alone.
>
> We agree with the reviewer that these results are very interesting to the community and we will highlight them more clearly in the main body. For now, however, we updated the discussion in Section 5.4 to account for our new results.
>
> >Some analyses I would like to see in this regard: a comparison of training on synthetic data from model 1, but an evaluation with another, independent structure prediction model 2.
>
> As mentioned earlier, we already provide a similar experiment with our initial submission. Furthermore, we now include evaluations of 3D structures using AlphaFold 3 predictions (Tabels 22 and 23) as explained above.

---

> ### Author Response · Authors · 2024-11-25
> **Author response to Reviewer DDzf continued**
>
> >an analysis of the sequence recovery (an imperfect metric, I know, but still somewhat informative to exclude cases of the inverse folding model overfitting on quirks of the structure prediction model). If you were to go all out, an evolutionary inspired recovery may be even more informative (c.f. https://openreview.net/forum?id=y5L8W0KRUX&referrer=%5Bthe%20profile%20of%20Chengyue%20Gong%5D(%2Fprofile%3Fid%3D~Chengyue_Gong1) in the protein world
>
> We thank the reviewer for this useful comment and the interesting reference. We have added the sequence recovery metric to the revised version of our manuscript.
>
> >In Table 1, it'd be helpful if the authors could highlight what % means (here % solved), as well as the n=... that was used for this estimate and the topk from which the answer was picked. From the description, it is not quite clear to me how those % were obtained.
>
> We thank the reviewer for pointing out this issue. We updated the table in the revised version of the manuscript to avoid confusion.
>
> >Were all benchmarked methods re-trained on the same synthetic data? (for each of the tasks)
>
> For our experiments, only the LEARNA family of algorithms (LEARNA, Meta-LEARNA, Meta-LEARNA-Adapt, libLEARNA) are learning based approaches. These, however, are automated reinforcement learning approaches that do not directly provide training pipelines, but rather, these methods employ large scale joint architecture and hyperparameter search to directly evolve a trained architecture with corresponding hyperparameters. Running these pipelines requires substantial computational resources and we did not rerun this optimization procedure.
>
> We would like to thank the reviewer again for the valuable feedback. We hope that we addressed all the questions and concerns satisfactorily and would appreciate it if the reviewer would consider increasing our score.
>
> With kind regards,
>
> The Authors

---

### Official Review · Reviewer_c4ig · 2024-11-09

**Soundness:** 2
**Presentation:** 2
**Contribution:** 2
**Rating:** 5
**Confidence:** 3

**Summary:**

This paper proposes RNAinformer, a novel generative transformer based approach to the inverse RNA folding problem. Leveraging axial-attention, they model the secondary structure input represented as an adjacency matrix in a 2D latent space, which allows us to invert all existing secondary structure prediction algorithms. The authors claim that RNAinformer is the first model capable of designing RNAs from secondary structures with all base interactions, including non-canonical base pairs and tertiary interactions like pseudoknots and base multiplets.

**Strengths:**

This paper studies RNA inverse folding (from secondary structures), which is an important problem in biology

**Weaknesses:**

* The methodology is a standard transformer and lacks innovation

**Questions:**

* Have you tried finetuning RNAinformer using the data from PDB? You said secondary structures derived from PDB are the golden standard but the dataset size is small. That's why you used synthetic data to pre-train. I wonder if it can improve your model performance with additional finetuning.

---

> ### Author Response · Authors · 2024-11-25
> **Author response to Reviewer c4ig**
>
> Dear Reviewer c4ig,
>
> We thank you for your valuable feedback and for pointing out the importance of our approach for the field of biology. We will address your concerns and questions in detail in the following.
>
> >The methodology is a standard transformer and lacks innovation
>
> We agree with the reviewer that our approach employs rather standard deep learning techniques including axial attention and auto-regressive generation. However, we would like to emphasize that the major novelty of our approach is about the application of these methods to a long-standing problem of computational biology. In this regard, the usage of axial-attention to process an RNA structure represented as an adjacency matrix is novel, offering multiple advantages over existing secondary structure-based RNA design algorithms as discussed in the Introduction of our initial submission. Most importantly, the right combination of these ‘standard’ techniques enables RNA design for nucleotide interactions that were previously intractable with other secondary structure-based RNA design approaches.
>
> >Have you tried finetuning RNAinformer using the data from PDB? You said secondary structures derived from PDB are the golden standard but the dataset size is small. That's why you used synthetic data to pre-train. I wonder if it can improve your model performance with additional finetuning.
>
> We thank the reviewer for this helpful comment.
> We pre-trained a RNAinformer model on synthetic data and data from known RNAs, and finetuned both models on the PDB data.
> The results are shown in Table 19 in Appendix E.4 in the revised version of our manuscript.
>
> We observe that finetuning indeed improved performance on the different PDB test sets. To further investigate this, we also evaluated the models on RNA only data from the CASP15 competition (results shown in Table 20 and 24 in Appendix E.4). Here, we observe that the finetuned model performs slightly worse than the model trained only on synthetic data. We conclude that finetuning can be beneficial in specific cases but that training on synthetic data seems to be beneficial.
>
> We again thank the reviewer for the useful feedback. We hope that we addressed all questions and concerns but would be happy to answer further questions if necessary.
> If there are no further questions, we would like to kindly ask the reviewer to consider increasing our score.
>
> With kind regards,
>
> The Authors

---

### Author Response · Authors · 2024-11-15
**Initial author response**

We thank all reviewers for their useful comments and valuable feedback.

To reduce the overhead for the reviewers, we will prepare individual responses for each review in the next few days.

We are looking forward to fruitful discussions and an interesting rebuttal period.

Best regards,

The authors

---

### Author Response · Authors · 2024-11-25
**Changes to the manuscript**

We thank all reviewers for their patience and we apologize for the late response. We have uploaded a revised version of our manuscript. The key changes we would like to highlight are the following:

- We add a section about 3D RNA design approaches to the related work in response to the comments of reviewers DDzf and VnQq.
- We implement a GNN baseline in response to the comments of reviewers Lp2j and VnQq. The results are shown in Table 16.
- We evaluate RNAinformer on the RNA only data from the CASP15 blind competition in response to the comment of reviewer VnQq. We use these evaluations to assess the influence of finetuning RNAinformer with experimentally validated samples from PDB as requested by reviewer c4ig. The results are shown in Table 20.
- Evaluations with AlphaFold 3 for 3D structure predictions
  - We evaluate the generated sequences using AlphaFold 3 in response to the comment of reviewer VnQq. We analyzed the result in terms of improved foldability as shown in Table 22. We found that, while we cannot improve the general foldability of the sequences, this is mainly due to a lack of MSA for the designed sequences. Therefore, we show the difference in TM score for examples where there was MSA available for the ground truth sequence and for orphan RNAs, highlighting that RNAinformer designs improve the foldability of orphan RNAs across nearly all test sets.
  - We also evaluate the predictions of RNAinformer on the CASP15 data with AlphaFold 3. The results are shown in Table 24 and indicate that the RNAinformer model finetuned on known samples seems to achieve worse performance compared to the version trained only on synthetic data.
- We deepen the analysis of training on synthetic data by folding all predictions of the RNAinformer version trained on known RNAs with different folding algorithms as requested by reviewer DDzf. The results are shown in Table 21.
- We extend the comparison of training on synthetic data, synthetic training with finetuning on experimentally validated structures from PDB, and training on known RNAs with finetuning on PDB samples as requested by reviewer c4ig. The results are shown in Table 19.

Any issues with rendering of Figures and text above the page limit in the current intermediate version of our manuscript will be resolved in later versions.

We also post individual responses to all reviewers. We are looking forward to a fruitful discussion.

With kind regards,

The authors

---

### Meta-Review · Area_Chair_Pbd2 · 2025-01-02

**Metareview:**

The paper gives a generative transformer model for the inverse RNA folding problem (designing RNAs from secondary structures). The problem that the paper studies is undoubtedly important, and the pipeline for synthetic data generation that the authors have developed can have other uses. However, because the paper is applying an established methodology rather than developing a new one, the bar for experimental evaluation is high, and the paper falls short by this measure. Specifically, the paper misses some important comparisons with other relevant prior efforts -- in particular, generation from RNA tertiary structures as well as efforts on protein-folding. The concerns are substantial enough that I must recommend rejection this time around. I encourage the authors to incorporate the feedback in the reviews and submit to a different deadline.

**Additional Comments On Reviewer Discussion:**

There was significant discussion between the authors and the reviewers during the rebuttal period. In the end, the authors were unable to convince the reviewers.

---

### Decision · Program_Chairs · 2025-01-22

Reject